# Genomes of fungi and relatives reveal delayed loss of ancestral gene families and evolution of key fungal traits

Zsolt Merényi ●[1], Krisztina Krizsán[1,4], Neha Sahu[1], Xiao-Bin Liu[1], Balázs Bálint ●[1], Jason E. Stajich ●[2,3], Joseph W. Spatafora ●[2] & László G. Nagy ●[1]✉

Fungi are ecologically important heterotrophs that have radiated into most niches on Earth and fulfil key ecological services. Despite intense interest in their origins, major genomic trends of their evolutionary route from a unicellular opisthokont ancestor to derived multicellular fungi remain poorly known. Here we provide a highly resolved genome-wide catalogue of gene family changes across fungal evolution inferred from the genomes of 123 fungi and relatives. We show that a dominant trend in early fungal evolution has been the gradual shedding of protist genes and the punctuated emergence of innovation by two main gene duplication events. We find that the gene content of non-Dikarya fungi resembles that of unicellular opisthokonts in many respects, owing to the conservation of protist genes in their genomes. The most rapidly duplicating gene groups included extracellular proteins and transcription factors, as well as ones linked to the coordination of nutrient uptake with growth, highlighting the transition to a sessile osmotrophic feeding strategy and subsequent lifestyle evolution as important elements of early fungal history. These results suggest that the genomes of pre-fungal ancestors evolved into the typical filamentous fungal genome by a combination of gradual gene loss, turnover and several large duplication events rather than by abrupt changes. Consequently, the taxonomically defined Fungi represents a genomically non-uniform assemblage of species.

The evolutionary diversification of lineages into clades of various sizes and phenotypes, some highly successful and species-rich while others less so, is an outcome of complex interactions between ecological opportunity, changing biotic and abiotic environments and the genetic make-up of the organisms representing the lineage. Inferring the genomic footprint of the emergence of high-ranking clades and innovations that underlie their evolutionary success have been hard to assess. Recently, comparative genomic approaches that allow ancestral genome content to be inferred from extant genomes have revealed that complex mechanisms, including gene birth, loss and co-option[1–3], contributed to the origins of high-ranking clades, including prokaryotes[4] and eukaryotes[5], metazoans[1,6] or land plants (embryophytes[2]). These patterns are consistent with proposed general trends of genome evolution[7]; however, whether they are universal across the tree of life is unknown.

Fungi are one of the evolutionarily most successful groups. They exhibit extreme diversity in both morphology and ecological

[1]Synthetic and Systems Biology Unit, Institute of Biochemistry, Biological Research Centre, Szeged, Hungary. [2]Department of Botany and Plant Pathology, Oregon State University, Corvallis, OR, USA. [3]Department of Plant Pathology and Microbiology, University of California, Riverside, CA, USA. [4]Present address: Institute of Forensic Genetics, Hungarian Institute for Forensic Sciences, Budapest, Hungary. ✉e-mail: lnagy@fungenomelab.com

function and play key roles in many ecosystems as symbionts, parasites and saprobes, among others. The most typical manifestations of the fungal morphology, that is, non-motile, septate, filamentous thalli, are found in the Dikarya, but taxonomically, Fungi represent a much broader spectrum. Fungi belong to the Holomycota in the Opisthokonta, together with the Nucleariida[8]. Defined in the broad sense, Fungi encompass at least ten phylum-level clades, with most of their phylogenetic diversity found in non-Dikarya fungi (also known as early-diverging fungi). This is a paraphyletic group comprising the Chytridiomycota, Sanchytriomycota, Blastocladiomycota, Olpidiomycota, Zoopagomycota and Mucoromycota as well as Aphelida[9] and the Rozellida + Microsporidia clade (Opisthophagea[10])[11–14]. Notwithstanding the continuous flux in the taxonomic definition of Fungi[15], the phenotypes of the earliest fungal ancestors are characterized and how they evolved from a unicellular opisthokont ancestor has been the subject of intense research. Reconstructions of the last universal fungal ancestor (LUFA) became more and more nuanced with the inclusion of newly discovered and/or characterized early-diverging lineages[9,10,12,16–19]. These results contributed to our current understanding of the LUFA as a unicellular phagotrophic parasite of microalgae, possessing motile cell stages with amoeboid and flagellar motility and a chitinous cell wall, at least in part of its life cycle[10,11]. By contrast, derived fungi (that is, Dikarya) are sessile terrestrial osmotrophs that grow septate hyphae covered by a rigid cell wall throughout their life cycle. How the LUFA and subsequent ancestors evolved modern fungal traits is poorly known, and systematic analyses of the genomic changes during this transformation are missing.

In this study, we reconstructed the gene family evolution in the Holomycota using 123 whole genomes and analysed temporal and functional trends in the genomic changes we inferred. We found that non-Dikarya fungi are genetically intermediate between pre-fungal protists and the Dikarya and that a gradual shedding of ancient protist gene families happened in parallel with the emergence of novel families and the expansion of pre-existing families in Fungi. The tempo and mode of gene turnover and innovation outline major genomic trends and reveal an episodic emergence of modern fungal traits, including multiple waves of gene family expansion and contraction related to key fungal traits. Our results reveal that taxonomically defined fungi do not match with 'genomic fungi' and that early fungal evolution has been highly episodic with continuous gene turnover.

## Results and discussion

### Gene content of non-Dikarya fungi resembles that of protists
To obtain a global perspective on gene content differences between fungi and related opisthokonts, we first clustered the protein sequences of 123 species into homologous protein groups (HGs; Methods). For this, we selected representatives of all currently accepted Holomycota phyla except Sanchytriomycota, as well as 16 other Amorphea (including metazoans and single-celled representatives) and a Heterolobosea species as an outgroup (Supplementary Data 1). For short, unicellular non-fungal eukaryotes are referred to as protists hereafter. We inferred a species phylogeny by maximum likelihood analysis of a supermatrix of 272 single-copy orthologues (Fig. 1a and Extended Data Fig. 1). Our phylogeny is highly supported and is largely congruent with phylogenies from recent studies[10,11,20–22].

Based on the HG membership data, a principal coordinate analysis (PCoA) resolved phyla into distinct groups, indicating that gene content divergence correlates well with phylogeny (Fig. 1b and Extended Data Fig. 2). Non-Dikarya fungi, especially chytrids and Aphelida, grouped with non-fungal protists instead of occupying distinct positions in the space. The Blastocladiomycota and Zoopagomycota were transitional towards the Mucoromycota + Dikarya (Fig. 1b). The Dikarya branched into two groups in a tree-like pattern corresponding to the Ascomycota and Basidiomycota, with secondarily reduced yeast-like lineages of both phyla falling closer to each other. The intermediate placement

of non-Dikarya fungi in the genomic space is consistent with a number of phenotypic (amoeboids, flagellated zoospores), ecological (parasites, phagotrophic feeding), biochemical and genetic similarities[13,23]. Some of these have already been studied in detail, such as the distribution of flagellar genes[24], cytoskeletal complexity[25], class V–VII chitin synthases[24], the cell-cycle system[26], the Wiskott–Aldrich syndrome protein (WASP) and SCAR/WAVE homologues[17] or cobalamin utilization[27]. However, whether protist genes are ubiquitous in non-Dikarya fungi has not been systematically investigated. On the basis of these observations, we asked whether the presence of protist-like genes is a broad genomic trend in non-Dikarya fungi and, if so, how these were replaced by fungal novelties during the evolution of the Holomycota.

### A protist genomic heritage and limited novelty in fungi
To identify genes lost during early fungal evolution, HGs showing >70% conservation in at least one taxonomic group were considered (Supplementary Data 1). We identified 540 HGs that must have been present in the Holomycota ancestor but are absent in Dikarya (Fig. 2 and Supplementary Data 2), indicating loss events. Of these, we inferred that the LUFA lost 41 HGs when it diverged from the nucleariids. The largest loss events, 76, 64 and 239 HGs, were inferred in nodes in which Blastocladiomycota, Zoopagomycota and Dikarya, respectively, split from their immediate ancestors. This pattern suggests that genes conserved in pre-fungal protists were lost in a step-wise manner, and non-Dikarya fungi possess a substantial number of HGs shared with non-fungal lineages, considerably more than what previous anecdotal evidence suggested.

The 540 HGs included all gene families previously reported to be lost in fungi: subunits of WASH and vasodilator-stimulated phosphoprotein (VASP) (lost in LUFA), WAVE[18], the CyclOP[28] light-sensing system (lost after Blastocladiomycota), flagellar genes[24] (lost across the branching of Chytridiomycota, Blastocladiomycota and *Olpidium*; Supplementary Data 3), cobalamin synthesis proteins[27] and the replacement of the E2F-type TF (E2F) cell cycle regulator by the Swi4-Swi6 cell cycle box binding factor complex[26]. However, previous evidence covered only a minority (73 HGs, 13.5%) of the 540 HGs and our improved taxon sampling pinpointed the placement of these loss events in several cases. To understand the broad functions of HGs lost during early fungal evolution, we performed Gene Ontology (GO) enrichment analyses (Fig. 2 and Supplementary Data 2). This revealed a significant overrepresentation of several GO terms (Fisher's exact test, P < 0.05, Supplementary Data 2), for example, in relation to $Ca^{2+}$-binding proteins and members of inositol 1,4,5-trisphosphate (IP3), diacylglycerol (DAG) and $Ca^{2+}$ signalling pathways. In these pathways, we inferred both a copy number reduction and complete loss of HGs (Extended Data Figs. 3 and 4). The shared ancestry of $Ca^{2+}$ signalling in Fungi and Metazoa was known[29], but our results revealed that the full complement of these pathways has been retained in fungi until the Mucoromycota–Dikarya split. Our dataset also showed that the gamma-secretase complex, which is involved in regulated intramembrane proteolysis for various developmental and signalling processes and was thought to be absent in fungi[30], in fact, was lost only in Dikarya, Olpidium and the Rozellida-Microsporidia (RM clade). Interestingly, several HGs associated with ubiquitination have also been lost during fungal evolution. Other functions highlighted by GO correspond to families annotated in extant Holozoa as mechano- and voltage-sensitive channels, receptors (epidermal growth factor, G-protein-coupled receptors (GPCRs)) and mechanical or visual perception (Supplementary Data 2). We have relied here on metazoan GO annotations, as these are the most complete, but note that they have limitations, as they might provide little insight into the functions of families in the pre-fungal protists and non-Dikarya fungi.

We also looked at core fungal novelties, defined here as HGs that evolved in one of the early fungal ancestors and are highly conserved (≥70%) in descendant lineages. Previous studies have reported a

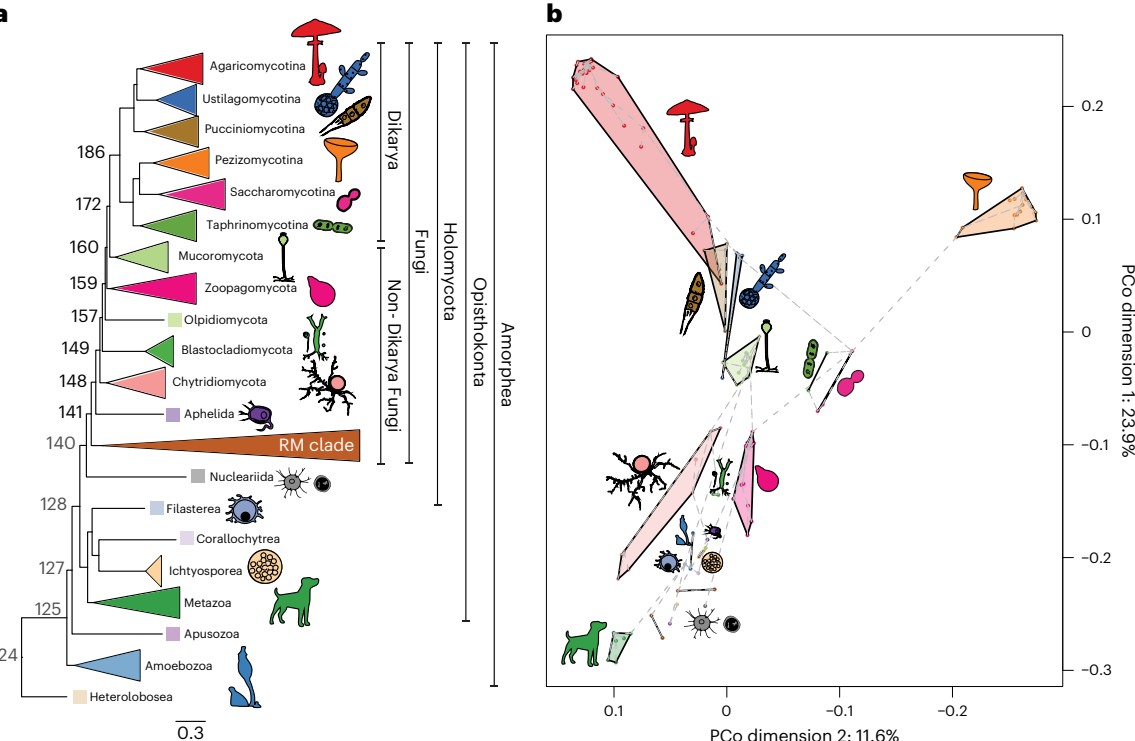

**Fig. 1 | Intermediate position of non-Dikarya fungi between protists and Dikarya. a**, Schematic representation of our species tree showing major clades and node numbers. The bold numbers at the nodes denote common ancestors of fungal phyla. See Extended Data Fig. 1 for the complete species tree and support values. Scale bar indicates number of substitutions per site. **b**, PCoA and minimum spanning tree (dashed grey line) based on presence and absence data of 9,993 HGs. Microsporidia species were removed from this analysis for better visualization.

shortage of fungal synapomorphies[15], so we tested whether our genome-wide dataset can provide a different viewpoint. We identified 163 HGs, considerably less than the number identified in recent studies, in animals and green plants (using a higher, 95%, conservation threshold[1]). This suggests that novel core HGs are less prevalent in fungi, possibly because fungi comprise several highly reduced clades (for example, yeast lineages[31] and Microsporidia). The 163 families originated across multiple nodes, with a larger grouping at the split of the Chytridiomycota from other fungi ($n = 57$; 35%, node C; Fig. 2), suggesting that this node has seen key transitions in genome evolution (see below). Conserved domain analysis of these 163 HGs revealed that 72 contain protein domains that are predominantly (≥99%) found in fungi, corroborating them as real fungal novelties. These have functions in spore formation (for example, Spo71 and Suppressor of *rvs167* mutation (Sur7) of *Saccharomyces cerevisiae*), mating (for example, Ste3, Prm1 and Rsc7/Swp82 of the SWItch/Sucrose Non-Fermentable (SWI/SNF) complex of *S. cerevisiae*) or cell polarity (for example, Spa2/Sph1 and SOG2 of *S. cerevisiae*), among others (Supplementary Data 4a). We also detected families related to the cytoskeleton, the fungal cell wall (FCW; for example, the Kre9/Knh1 family), intracellular trafficking, transporters and fungal-type mitosis and meiosis. We found a significant enrichment of transcription factors among novel HGs (Fisher's exact test, $P = 0.002$), including the origins of APSES, Copper fist, Opi1 and Fungal trans 2 families (see details below). Notably, the Velvet, Gti1/Pac2 and STE-like transcription factor (TF) families, which are thought to be fungal specific, had representatives in early-diverging holozoans (*Capsaspora*, *Salpingoeca* and *Corallochytrium*) or in *Fonticula*. Using a domain-based search logic, we further identified 186 less conserved HGs comprising fungal-specific domains, with functions relevant to sporulation, mating, intracellular transport, environmental sensing and chromatin remodelling, among others (Supplementary Data 4b).

Finally, HGs containing unannotated proteins are prevalent (17.8% of 163 and 14.5% of 186 HGs) among core fungal novelties, highlighting the understudied status of fungal-specific genes (Supplementary Data 4ab).

Taken together, our analyses revealed that non-Dikarya fungi possess a large number of HGs shared specifically with protists and that these were gradually lost during evolution. The broad conservation of protist HGs may explain the morphological and genetic similarities between non-Dikarya fungi and protists reported in previous studies[19,25,26,32,33] and also clarifies why the genome content of fungi reflects that of ancestral opisthokonts more than the metazoan gene content does[34]. Our data revealed that the retention of protist genes is not restricted to certain HGs but is a genome-wide trend in non-Dikarya fungi. At the same time, we identified several conserved homologous groups that originated in early fungi and were mostly conserved afterwards, as well as 186 less conserved families containing fungal-specific domains. Although most of these cannot be considered synapomorphies in the strict sense[15] because of their incomplete conservation, this indicates that beyond losses, considerable novelty has also emerged in early fungal ancestors.

### An interplay of gene gain and loss shaped fungal evolution

To obtain a detailed picture of gene repertoire changes and to test whether evolution is gradual or rather episodic, as some theories predict[35], we reconstructed the gene gain and loss dynamics for all HGs containing at least four proteins. Our reconstructions provide information on which genes were duplicated and lost in each of the HGs and at which branches of the phylogenetic tree (Fig. 3). For example, the LUFA was inferred to have had 12,761 genes, gained 913 and lost 295 compared with its immediate ancestor, corresponding to a moderate net expansion. Reconstructed ancestral proteome sizes ranged from

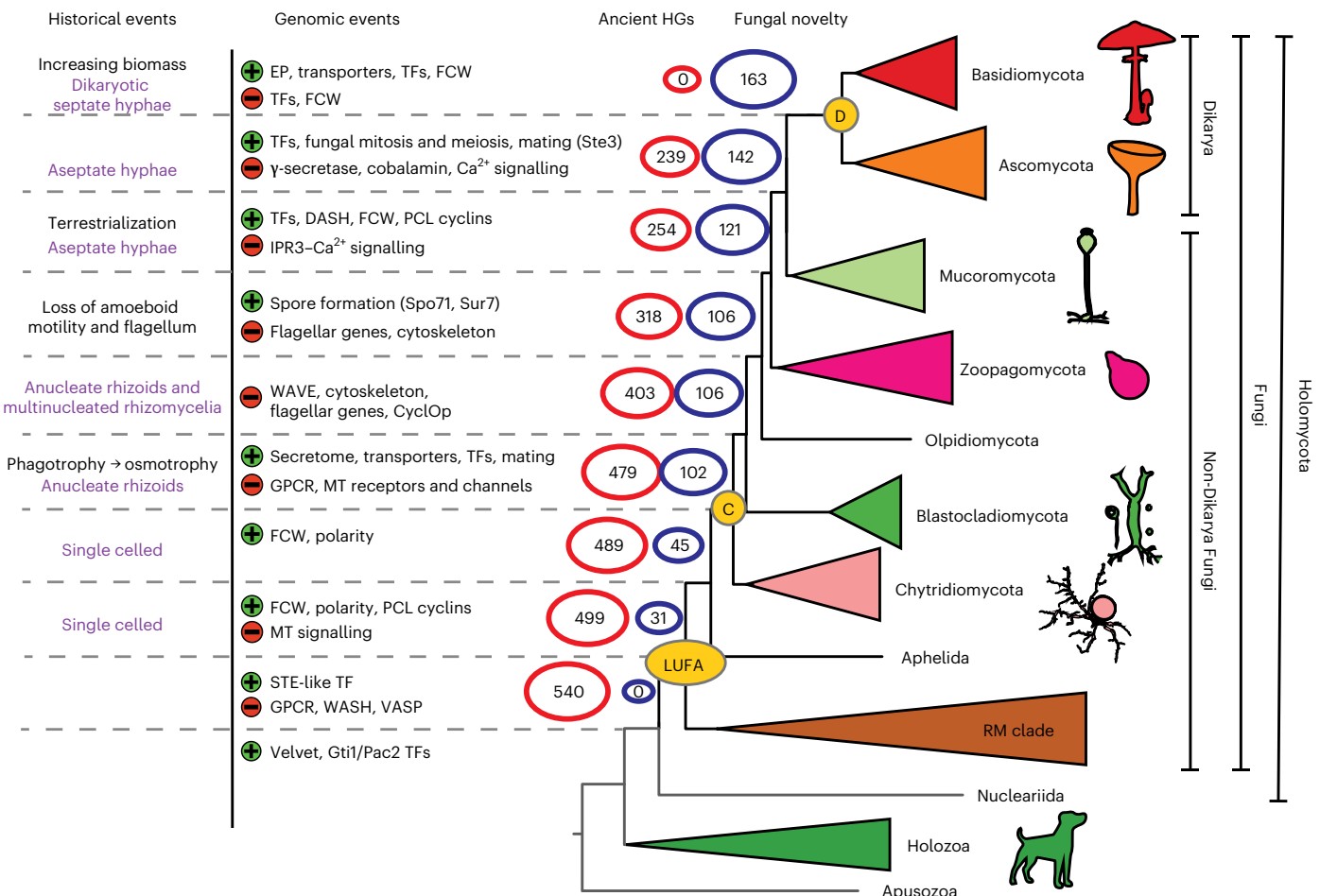

**Fig. 2 | Emergence of fungal novelties and loss of protist-conserved HGs during early fungal evolution.** Historical events[13,50], probable cellular complexity (purple text) and noteworthy genomic events (green, '+', gain or expansion; red, '−', loss or contraction) that we detected are shown at corresponding nodes. Numbers at nodes indicate the number of protist-conserved (red) and fungal-specific (blue) HGs. C, MRCA of Chytridiomycota and derived fungi; D, MRCA of Dikarya; EP, extracellular proteins; DASH, fungal-specific part of the kinetochore complex; MT signalling, metazoa-type signalling (for example, $Ca^{2+}$ signalling).

12,761 protein-coding genes in the LUFA to 14,891 in the most recent common ancestor (MRCA) of Zoopagomycota and derived fungi. If we consider net changes (duplications minus losses), it seems that most of the genomes of the early fungal ancestors contracted (Fig. 3), with two exceptions. The first is the split of chytrids from other fungi, and the second is the MRCA of Zoopagomycota and derived fungi, which is inferred to have expanded by 791 genes (1,114 gains, 323 losses).

We find that gene duplication has been highly episodic, with five large bursts, whereas losses were ubiquitous across the dataset. Of the five bursts of duplication, two were inferred in ancestral fungal nodes (nodes C and D in Fig. 3), two in phylum-level clades (Neocallimastigomycota and Leotiomyceta) and one in the Opisthokonta MRCA. The duplication event in the MRCA of chytrids and other fungi (C) was associated with limited gene loss (2,455 gains, 382 losses), whereas the one in the Dikarya (D) coupled with more loss (2,875 gene gains, 3,036 losses), suggesting it is better characterized as a turnover event. In general, ancestral proteome sizes were relatively constant, which, in joint consideration with the losses and pulses of gene duplication, indicates high gene turnover in early fungal evolution. Thus, while proteome sizes changed moderately, gene content has undergone notable changes. These patterns are consistent with those reported in a recent study[34], both in terms of the relative constancy of the total proteome size and the proportion of gains and losses at the nodes, though the exact numbers of duplications and losses differ.

The chytrid and Dikarya duplication events contained a similar functional signal, with an enrichment of terms related to extracellular functions (for example, GO:0005576), transmembrane transport (for example, GO:0055085), cellulose binding (GO:0030248) and the FCW (for example, GO:0016977). These may correspond to the transition from phagotrophy to osmotrophy and the chitinous cell wall, respectively, and reflect the improvement of extracellular digestive functions, as adaptations to the existence of increasingly complex and abundant plant material[13,36]. A significant enrichment of TFs was detected among duplicated genes across multiple nodes (Fisher's exact test $P < 0.05$, Supplementary Data 5), consistent with the results of novel core families (see above). On the basis of GO results, we identified an expansion of PCL cyclins that transcends multiple nodes in early fungi (Extended Data Fig. 5 and Supplementary Data 5). In extant species, Pho85p cyclins (PCL) coordinate the cell cycle with polarized growth in response to nutritional cues (phosphate, amino acid and glycogen metabolism)[37,38]; thus, their diversification might have provided the basis for the evolution of the sophisticated coordination of nutrient supply and growth. The two bursts of duplication in the Leotiomyceta and in the Neocallimastigomycota may correspond to periods of intense lineage-specific innovation which, in the latter clade, probably also reflect massive gene gains through horizontal gene transfer from bacteria, as reported earlier[39,40].

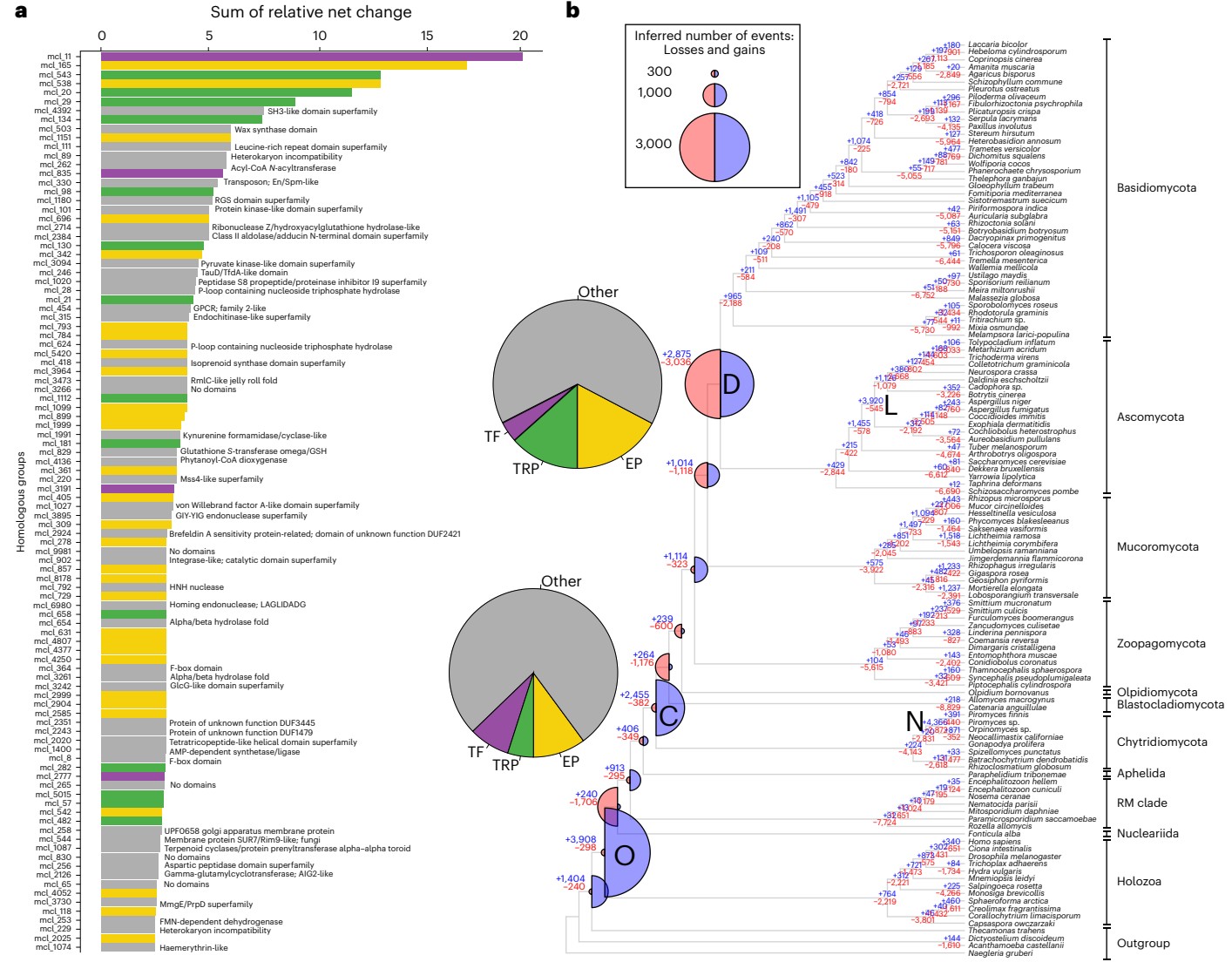

**Fig. 3 | A global view of gene gain, duplication and loss across fungal evolution. a**, The top 100 most dynamically changing HGs, coloured by functional categories (TRP, transporters). InterPro annotations are shown only for the 'Other' category. **b**, Gene duplication and loss history during fungal evolution. Blue (expansion) and red (contraction) half circles indicate the inferred gains and losses at each node, respectively, and blue (gain) and red (loss) numbers in nodes depict the number of gain and loss events. Note that the ratio of blue and red circles is proportional to the amount of net innovation and turnover in each node. Letters in bold represent the five largest bursts of duplication that comprised more than 2,000 gain events. Pie charts, next to nodes D and C, represent the contribution of four functional categories to gene gains. Clade names are abbreviated as O, Opisthokonta; C, MRCA of Chytridiomycota and derived fungi; D, MRCA of Dikarya; N, Neocallimastigomycotina; L, Leotiomyceta.

Data for the first two nodes are not shown as they mostly contain pan-eukaryotic genes. Acyl-CoA, acetyl coenzyme A; AIG2, protein of *Arabidopsis thaliana*; AMP, adenosine monophosphate; DUF1479, domain of unknown function 1479; DUF2421, domain of unknown function 2421; DUF3445, domain of unknown function 3445; En/Spm, Suppressor-mutator elements; F-box domain, PF00646; GlcG, GlcG gene of *Escherichia coli*; GSH, glutation; LAGLIDADG, sequence motifs for this DNA endonuclease; MmgE/PrpD, 2-methylcitrate dehydratases and FMN, flavin mononucleotide. Mss4, mammalian suppressor of Sec4; RGS, regulator of G protein signalling; Rim9, pH-response regulator protein of *S. cerevisiae*; RmlC, deoxythymidine diphosphates-4-dehydrorhamnose 3,5-epimerase; SH3, src Homology-3; TauD/TfdA, taurine dioxygenase/alpha-ketoglutarate-dependent 2;4-dichlorophenoxyacetate dioxygenase; UPF0658, Golgi apparatus membrane protein.

## The impact of genomic changes in trait evolution

We were next interested whether the functional profile of inferred bursts of duplication corresponds to hypothesized phenotypic changes in early fungal ancestors[13]. The most dynamically changing (expanding and contracting) homologous groups and corresponding functions were identified by ranking families by their summed expansion and contraction dynamics across eight nodes from the LUFA to the MRCA of Dikarya (Supplementary Data 6). Among the 100 most dynamically expanding families, 51 contained extracellular proteins (mostly carbohydrate-active enzymes, briefly CAZymes), transporters or transcription factors (Fig. 3a). The remaining families

included diverse functions, such as GPCRs, heterokaryon incompatibility genes and protein kinases, among others (Supplementary Data 6). When we parsed these figures in the context of the large duplication events in early fungal ancestors, we found that 22.9% of the chytrid and 34.7% of the Dikarya duplications were related to extracellular functions, transporters and transcription factors (Fig. 3b). The importance of these HGs for genomic changes was also confirmed by the functional enrichment analyses of HG duplication data (Supplementary Data 5). On the basis of these observations, we scrutinize below the extracellular, transporter and transcription factor families in more detail.

Mapping of families with predicted extracellular localization revealed two-stage duplication dynamics, with large expansions in the MRCA of chytrids and the rest of the fungi (from 441 to 653 genes) and in that of Dikarya (from 806 to 1,167 genes; Extended Data Fig. 6). The chytrid expansion was driven mainly by CAZymes, whereas other extracellular functions, including those of proteases, made a more modest contribution. Overall, owing to these expansions, the CAZyme and the small secreted protein content of the secretome shifted from 34% in the LUFA to 51% in the Dikarya MRCA. Within extracellular proteins, plant cell wall-degrading enzymes (PCWDE) follow similar patterns, but with a larger expansion in the Dikarya (Extended Data Fig. 7). By contrast to PCWDEs, FCW-related families showed proportionally more diversification in the chytrids than in the Dikarya, and a large turnover in the Dikarya (Extended Data Fig. 8). The diversification of extracellular proteins, especially PCWDEs, in the ancestor of chytrids and other fungi, correlates with the transition from phagotrophy to osmotrophy, whereas the expansion of FCW-related families here may have contributed to the evolution of the FCW. The second burst, in the Dikarya, may be concomitant with the radiation of plant lignocellulose-degrading lineages, possibly in response to the onset of the radiation of land plants[36]. However, in the ancestors of Blastocladiomycota, Olpidiomycota and Zoopagomycota and other fungi, the lack of PWCDE expansion and a moderate expansion of proteases may be explained with the primarily non-plant-based nutrition of these clades[41].

For cell surface transmembrane transporters, we identified 253 HGs based on characteristic domains and subcellular localization (Supplementary Data 7). Similar to CAZymes, transporters show a two-stage expansion, with the highest number of duplications inferred in the Dikarya MRCA (Extended Data Fig. 9). Of the three largest transporter families, we found that adenosine triphosphate binding cassette transporters and P-type ATPases showed contraction, whereas the major facilitator superfamily underwent an extreme expansion in fungi. This makes sense considering that only the latter is able to transport a high variety of small molecules, including sugars, peptides and lipids[42], whereas adenosine triphosphate binding cassette transporters and P-type ATPases are primarily exporters[43] and specific for cations[44], respectively. Given the role of transporters in osmotrophic nutrition[45], the correlated diversification of transporters with CAZymes and the inferred continuous increase of copy numbers in ancient fungal genomes suggest their importance in the refinement of fungal-type heterotrophy. We identified 657 TF HGs based on domain content and classified them into 52 putative TF families (Supplementary Data 8) following previous studies[46,47]. Mapping of these 657 HGs revealed a constant change in TFome (general TF repertoire) during early fungal evolution. Early nodes, such as the splits of chytrids (193 gains, 3 losses) and Zoopagomycota (168 gains, 5 losses), are almost exclusively dominated by gains with barely any losses (Fig. 4). This implies that the TFomes of these clades are similar to those of related protists, as suggested in a previous study[47]. By contrast, later evolution of non-Dikarya fungi is characterized by a high turnover of TFs: at the MRCA of Mucoromycota and derived fungi, we inferred 87 gains and 43 losses, whereas in the Dikarya MRCA, we detected 117 gains and 184 losses. Losses in the Dikarya MRCA affected several families (for example, bZIP, Myb, CBF-NFYA, GATA, Homeobox) and included the complete loss of the E2F TDP, T-Box and Tub families. From the LUFA to the Dikarya, TFome diversity (based on the Shannon index) and the number of ancestral TFs increased with the peak inferred in the MRCA of Mucoromycota and Dikarya (784 TFs; Fig. 4a). In line with this, extant Mucoromycota contain the most TFs among fungi, with Shannon-based diversities similar to those of other non-Dikarya fungi (Fig. 4c). The TFomes of extant Dikarya, especially Ascomycota species, show similar sizes but lower diversity partly owing to the expansion of the fungal trans 2 and Zn cluster TF families and the losses of certain families.

The inferred copy number of TFs in the fungal ancestors outlined three 'epochs', within each of which ancestral TF repertoires seemed to be relatively constant (see shading in Fig. 4a). Transitions between these epochs are concomitant with remarkable historical events, such as the emergence of mostly anucleate rhizoids and osmotrophy or that of terrestrialization and of aseptate hyphae (Figs. 2 and 4a).

The ranking of TF families based on the cumulative net change during early fungal evolution (from the LUFA to the Dikarya; Supplementary Data 8) revealed that the most dynamically changing TF families included both pan-eukaryotic (for example, C2H2-like, bZIP, HLH, HSF, Homeobox and GATA) and predominantly fungal families (for example, Zn cluster, Fungal trans 2, APSES, Velvet and Gti1/Pac2). Several families expanded considerably more than the whole proteome, indicating that their copy numbers are decoupled from proteome size. For example, while the proteome size of the MRCA of chytrids and derived fungi increased by 1.16× (from 12,818 to 14,891 genes) relative to the preceding node, the Zn cluster and bZIP families expanded by 7.38× and 3.25×, respectively. Interestingly, C2H2 TFs underwent a massive expansion in the Opisthokonta ancestor. This family is highly diverse in both the Holozoa and the Holomycota (Extended Data Fig. 10).

Taken together, functional analyses of highly expanding homologous groups revealed processes that dominated duplications during early fungal evolution. Of these, the expansion of extracellularly secreted proteins and transcription factors may have facilitated the evolution of osmotrophy and that of fungal gene regulatory systems, respectively. Osmotrophy evolved independently in Fungi (Phytophagea[10]), Microsporidia (Opisthophagea) and the distantly related Oomycetes (Stramenopila[48,49]). It is noteworthy that while in fungi and oomycetes the evolution of osmotrophy correlates with gene gains, mostly transporters and extracellular digestive enzymes[48,49], in the Microsporidia, it correlates with genome reduction, possibly because the latter are intracellular parasites. The evolutionary dynamics of transcription factors differs from genome-wide patterns, in that, albeit also episodic, it showed expansions in different nodes. Overall, patterns of transcription factor evolution suggest that broad rewiring of fungal gene regulatory networks has transcended multiple ancestors in non-Dikarya fungi.

## Conclusions

The origins of highly diverse clades across the tree of life are noteworthy evolutionary events, but are they remarkable from a genomic perspective or because we attach taxonomic definitions to them? On the basis of systematic analyses, we inferred that protein coding gene content has changed drastically during early fungal evolution. However, in contrast to animals[3] and plants[2], this has not happened abruptly at the taxonomic limit of Fungi, even if competing taxonomic circumscriptions are considered. Rather, we found a remarkable retention and step-wise loss of protist genes in fungi, combined with the episodic expansion of novel and ancient homologous groups. We also identified major functional trends and the most dynamically changing HGs, which allowed us to relate genomic changes to trait evolution during the early evolution of fungi.

Our analyses revealed that non-Dikarya fungi retained hundreds of protist HGs that are missing in the Dikarya. Remarkable lost or contracting gene groups were related to phagocytosis, the flagellum, cell cycle regulation and signalling (Fig. 3), among others. These were lost gradually or replaced by fungal-specific genes (for example, E2F cell cycle regulator by SBF[26]) during fungal evolution. We think that the presence of these genes in non-Dikarya explains why they gravitate towards unicellular opisthokonts in gene content-based analyses (Fig. 1). They also provide a genome-wide explanation for previous anecdotal observations of similarity between non-Dikarya fungi and non-fungal opisthokonts, from analyses of single genes[26,27], individual genomes[34] or TFomes[47]. At the same time, we detected a limited number of HGs that could be considered synapomorphic for Fungi, in agreement with previous conjectures on the lack of fungal synapomorphies[15,24].

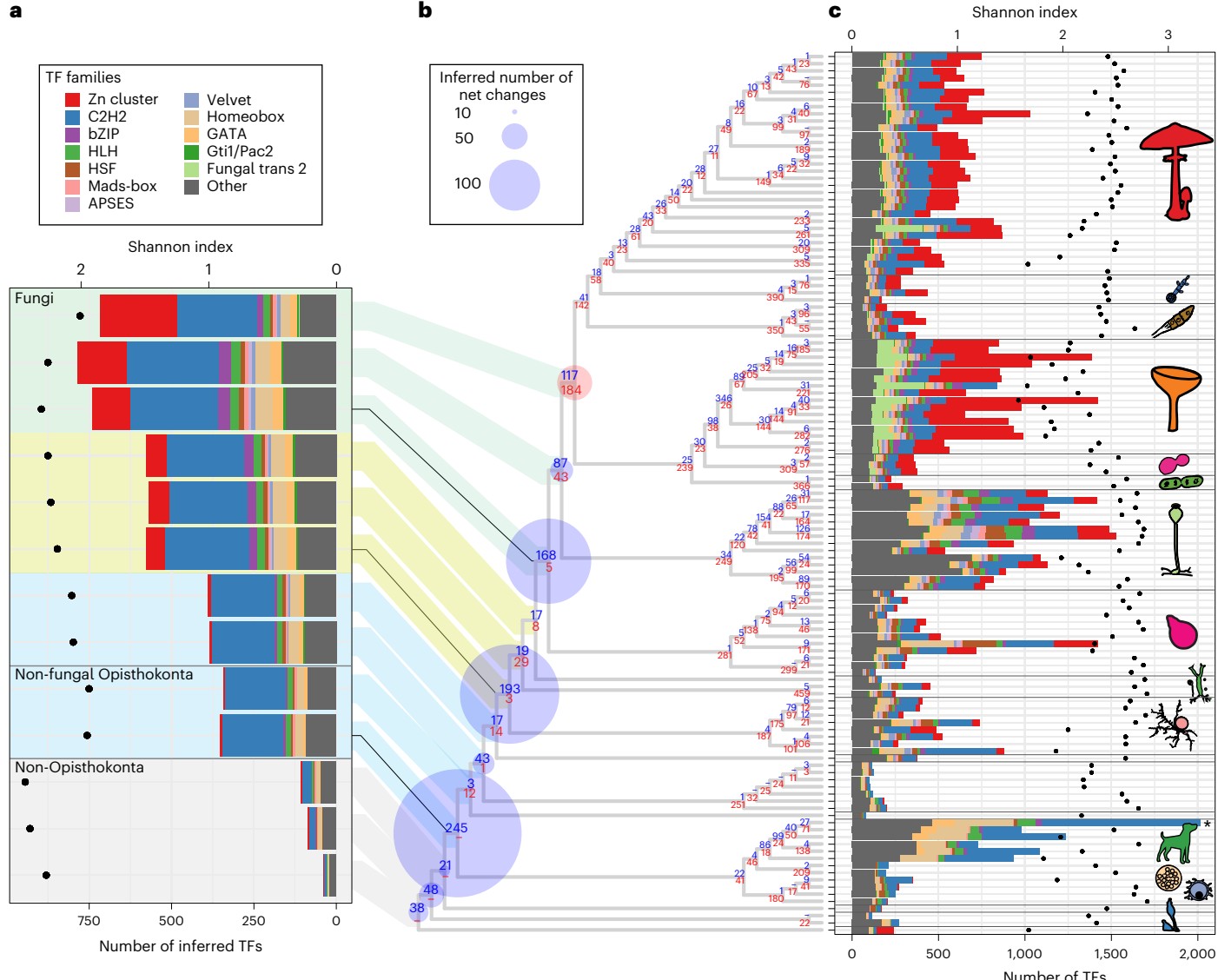

**Fig. 4 | Dynamic turnover of transcription factor families. a,** Inferred ancestral size of transcription factor families. On the basis of the inferred TFome sizes, three epochs were defined and highlighted with different background colours (blue, yellow and green). Only the 12 families that showed the highest dynamics in early fungal evolution were coloured differently (see main text and Supplementary Data 8). The Shannon index of TFome diversity for inferred ancestral species is shown with dots. **b,** Inferred net changes in TFs mapped to the species tree. Blue (expansion) and red (contraction) circles indicate the inferred net changes (gains minus losses) at each node, and blue (gain) and red (loss) numbers in each node depict the number of gain and loss events. **c,** The stacked bar chart represents the size of the TF families, using the same colouring scheme as in **a.** For visualization, the *Homo sapiens* column (asterisk) was scaled down proportionally from 5,262. The Shannon index of TFome diversity for extant species is shown with dots.

The retention of protist genes and the lack of clear synapomorphies for fungi blur lines between fungi and related protists and may explain why the taxonomic limits of fungi have been challenging to define and are still a matter of debate[18]. In other words, taxonomically defined Fungi do not match with any clade we could define, based on gene content, as 'genomic Fungi'. This situation would not change even with the exclusion of Opisthophagea from the fungal kingdom. Defining 'genomic Fungi' is also complicated by patterns of gene turnover. Nevertheless, gene content (Fig. 1), turnover rates (Fig. 2) and novel HGs suggest that, of any fungal clade, the Dikarya comprises species with a broader set of signature HGs. At the same time, the ancestor of the Chytridiomycota and derived fungi showed the second highest number of genomic innovations (novel HGs, Fig. 1; gene duplications: Fig. 3), propounding this node as a potential genomics-informed border of Fungi.

In contrast to gene loss, our inferences show that gene duplications were highly episodic, with large congregations of duplication in a few nodes of the fungal tree. Functional analysis of gene duplications outlined major functional trends in the evolution of early fungi. A dominant functional signal was related to extracellular proteins (for example, degradative enzymes) and transporters, which can be linked to the transition from phagotrophy to osmotrophy, and the subsequent sophistication of extracellular digestive and uptake mechanisms[10,13,15,18,41,50]. These also explained a considerable portion of the duplication events in the chytrid and Dikarya ancestors, although it should be noted that a myriad of other functions were also represented among the most expanding families. For example, transcription factors emerged as one of the most dynamically changing gene groups, suggesting a broad rewiring of gene regulatory mechanisms in early fungi. Another type of dynamically changing regulator was PCL-type cyclins,

which link nutritional status with the cell cycle. The expansion of this family is perhaps related to the balancing of nutrient assimilation and growth, an important regulatory mechanism for hyphal osmotrophs. Our genome-wide catalogue of changes revealed a large array of other functions as well, many of which cannot be easily linked to phenotypes or ecological functions owing to the paucity of knowledge on gene function, but represent interesting targets for functional analyses.

Taken together, this study reconstructed the history of genomic change in fungi at unprecedented detail and provides a resource for further analyses of fungal genome evolution, also at smaller evolutionary timescales. We conclude that, although sharp genetic changes at the taxonomic border of fungi were not inferred, a large turnover of protistan genes and a gradual emergence of fungal novelties in early fungal evolution portray a clear genetic roadmap for the emergence of derived fungi from their opisthokont ancestor.

## Methods

### Dataset assembly and clustering of proteins
To evenly cover the Holomycota lineage, we sampled 106 fungal species, consisting of proteomes from all known phylum-level clades, with the exception of Sanchytriomycota[28], which were not published before our data collection. In addition, as outgroups, 17 non-fungal species were sampled to represent Heterolobosea, Amoebozoa, Holozoa and Nucleariids. In this study, for simplicity, we use the term 'protist' as a paraphyletic group of unicellular and non-fungal eukaryotes. Proteome sequences were downloaded from the JGI Genome Portal and NCBI/ Ensembl (before July 2021)[51,52]. All versus all similarity searches of the 123 proteomes were performed with MMSeqs2[53] using three iterations, with sensitivity set to 6.5, max-seqs set to 15,000, e-profile set to 0.001, a preliminary coverage threshold set to 0.2 and an $e$ value threshold set to $1 \times 10^{-4}$. We then performed an asymmetric coverage filtering (requiring 20% coverage for the longer and 80% for the shorter protein) and Markov clustering with an inflation parameter of 2.0 (ref. 54) as described previously[55]. After clustering, we removed contaminating proteins from homologous groups according to a previous study[56]. Furthermore, to achieve better completeness of clusters without increasing noise, we merged clusters based on similarity, using the all versus all output of MMSeqs and the results of the hidden Markov Model (HMM) search between the consensus sequence and HMM profiles of clusters. Based on the MMSeqs output file, a network of clusters was constructed. These networks were iteratively reduced by excluding the weakest nodes—sorted by the number of connections between the two clusters, normalized to cluster size—until the maximum diameter of a network was three. Of these cluster pairs, only those that achieved an $e$ value cut-off at least $1 \times 10^{-10}$ in the HMM search (http://hmmer.org/) with an asymmetric coverage of 75% and 20% (HMM profile and consensus, respectively) and whose match score reached at least 75% of the self-match were allowed to be merged. We called these merged filtered clusters of protein HGs.

### Species tree reconstruction
For species tree inference, marker genes were selected from four sources: (1) single-copy HGs based on the clustering mentioned above, (2) clusters that were single copy after eliminating terminal duplications from gene trees inferred for each cluster using a custom-made script (https://github.com/zsmerenyi/compaRe/blob/main/Terminaldupdet.zip), (3) a HMM-based search of BUSCO version 3.0.2 (ref. 57) and (4) a HMM of JGI 1,086 marker gene sets[58,59] (https://github.com/1KFG/Phylogenomics_HMMs/tree/master/HMM/JGI_1086) using HMMER 3.3.2 (ref. 60). For the latter two, only best hits were used for each species. Subsequently, clusters were removed in which the average distance of amino acid (AA) alignment was ≥1.5 (dist.ml under the WAG model)[61]. Also, we eliminated candidates containing potential ancestral paralogues by filtering out those presenting clusters with low phylogenetic relations at the first split of a hierarchical clustering

of AA distances (using the default parameter of the hclust function of package stats version 3.6.2). Multiple sequence alignments were inferred using PRANK v.170427 (ref. 62) and trimmed using TrimAL v.1.2 (ref. 63) (-strict). Trimmed alignments shorter than 60 AA residues and containing <30 species were discarded, leaving 272 single-copy clusters resulting in 68,662 sites that were finally used for tree reconstruction. Phylogenetic inference was performed under maximum likelihood in IQ-TREE v1.6.12 (ref. 64) with ultrafast bootstrap[65] (1,000 replicates) based on the partitioned dataset of 272 clusters using the substitution model LG + G. More complex models (LG + C60 + G) had no effect on the branching of early-diverging lineages. We applied a constrained tree topology ((Allma1, Olpbor1), Ganpr1, Partr) to separate Aphelida from Blastocladiomycota, which caused no significant change in log-likelihood values as assessed by the Shimodaira–Hasegawa test[66] (Δlikelihood = 21.8, $P = 0.356$).

### Inference of genome-wide duplication and loss history
For gene-tree reconstructions, homologous groups containing at least four proteins were aligned using the L-INS-I or auto (if the former was not applicable) algorithm MAFFT v7.313 (ref. 67) and trimmed with TrimAL (-gt 0.2). Gene trees were inferred in RAxMLHPC-PTHREADS-AVX2 8.2.12 under the PROTGAMMAWAG model, and we estimated branch robustness using the Shimodaira–Hasegawa-like support[68]. Rooting and gene-tree or species-tree reconciliation were performed with NOTUNG v2.9 (ref. 69) using an edge-weight threshold of 80.

Gene duplication and loss histories were inferred by mapping orthologous groups delimited on the basis of gene trees to the species tree using Dollo parsimony implemented in a modified version of COMPARE[31]. Detected gene gains could be de novo origination, duplication, horizontal gene transfer or the result of undetectable distant homology; however, in this study, we did not attempt to separate these events. A custom R script (https://github.com/zsmerenyi/compaRe) was used to visualize the mapping results utilizing functions of the phytools, ape, tidyr and phangorn packages[61,70–73].

### Annotation of homologous groups
For the evaluation of domain content and GO terms of HGs, an InterProScan 5.47–82.0 (ref. 74) analysis was performed on all 123 proteomes. A GO enrichment was carried out using Fisher's exact test with the weight01Fisher algorithm of the topGO Bioconductor module[75], and a $P < 0.05$ was considered significant. For the enrichment analysis of 540 homologous groups lost across fungal evolution, the most complete, the *Homo sapiens* GO list, was used as a reference. This approach has its limitations, as it is hardly informative for ancient lineages, but it is the most carefully and completely annotated GO reference available. For the GO enrichment analysis of HGs that underwent duplication or loss events in the ancestors of fungal phyla, all species were used and frequencies of orthologous groups were taken into account for each node.

A uniform rule was used for further annotation of HGs: a HG was assigned to a group if >50% of its proteins were annotated with the same type of annotation (for example, a specific domain, a small secreted protein (SSP) or extracellular localization). For TF family identification, sequence-specific DNA binding domains (DBDs) were selected based on literature mining. From the putative TF HGs (containing >50% of a given DBD), we filtered out those containing domains characteristic of non-TF families or processes such as ribonucleases, metallopeptidases, chromatin remodelling or splicing. Finally, we classified the HGs into TF families based on their DBD content, following previous studies[46,47].

Extracellular protein identification was based on subcellular localization prediction by WoLFPSORT 0.2 (ref. 76) using the 'fungi' option. Proteases, SSPs and CAZymes were predicted to further differentiate the extracellular protein-containing HGs. For proteases, the non-redundant database was downloaded from MEROPS (on

27 September 2022, from https://www.ebi.ac.uk/merops/download_list.shtml merops_scan.lib) and used as a query against the protein sequences of the 123 species in a BLAST search, with 20% bidirectional coverage and a $1 \times 10^{-5}$ $e$ value cut-off, keeping the best hit for each subject protein. The prediction of SSPs was performed using a modified version of the bioinformatics pipeline[77] as follows: proteins shorter than 300 amino acids were subjected to signal peptide prediction in SignalP (version 4.1 (ref. [78])), with those containing a transmembrane helix predicted by TMHMM version 2.0 (ref. [79]) excluded. For the prediction of CAZymes, a HMM search was performed with dbCAN2, using dbCAN-HMM profiles (https://bio.tools/dbcan CAN[80]) as queries. Subsequently, CAZymes were classified according to a previous study[81]. FCWs and PCWDEs were based on the classification of CAZy families in previous studies[82,83].

The detection of transporters was based on the presence of characteristic InterPro (IPR) domains according to a previous study[84] and plasma membrane localization, predicted by WoLFPSORT. HGs were considered transporters if they contained >50% of transporter-specific domains and if plasma membrane localization (score >15) was the most likely within the HG.

### Identification of conserved and dynamically changing HGs
To obtain a holistic picture of similarities and differences in the gene repertoire of the 123 species, a PCoA was performed. For PCoA, a total of 9,993 HGs comprising at least four proteins and reaching a conservation of ≥50% of species in any clade were used (see clades in Supplementary Data 1). A binary distance coefficient from the homologous group presence and absence data were used, and a minimum spanning tree was superimposed on the distribution of species on the two principal coordinates (using packages of stat, ape and tidyverse[71–73]).

To identify conserved HGs, we defined eight groups, Metazoa, Chytridiomycota, Zoopagomycota, Mucoromycota, Pezizomycotina and Agaricomycotina as well as the paraphyletic 'non-opisthokonta outgroups' and 'basal Holozoa' (Supplementary Data 1). The conservation of a HG was calculated as the proportion of the number of species with a protein present to the total number of species in each of these groups. We considered HGs to be shared between non-Dikarya fungi and protists if they had at least 70% conservation in any of the above groups or if the HG emerged in or before the first Holomycota node and was missing in Dikarya.

For identifying fungal core novelties, we searched for HGs that emerged after node 141 (LUFA) and had ≥70% conservation for all descendants of the node in which it emerged. This is similar to 'novel core' families investigated previously in animals[85] and plants[2]; however, we chose a conservation threshold of 70% owing to the large number of secondarily simplified lineages (for example, yeasts, Microsporidia) among fungi[86].

To validate fungal core novelties and losses, the distribution of InterPro domains among high-ranking taxonomic groups (Viruses, Bacteria, Archaea, Chromalveolata, Chromista, Excavata, Plantae, Protists, Holomycota, Holozoa) was examined. InterPro annotated proteins from UniProtKB, together with InterPro annotation (Release 2022_01), and the NCBI taxonomy database[87] (in May 2022) were downloaded. The distribution of the 37,834 IntePro domains among 10 taxonomic groups was calculated by normalizing the total counts for each domain, based on altogether 230,895,644 proteins. We used this dataset to assess the fungal dominance of a domain. For example, a domain was considered as fungal specific if 99% of uniprot hits come from Holomycota (Supplementary Data 4). Also, this proportion was used for mining HGs containing at least 75% of fungal-specific domain (Supplementary Data 4b).

To assess the dynamics of HG changes in the ancestors of fungal phyla, the relative net change to the copy number of the preceding node was calculated and summed across over eight nodes from the LUFA to the MRCA of Dikarya (Supplementary Data 6 and 8).

### Reporting summary
Further information on research design is available in the Nature Portfolio Reporting Summary linked to this article.

### Data availability
All data supporting the findings of this study are available within the main text, Methods and Supplementary Information and in the figshare repository (taxonomic versus genomic fungi[88]): https://figshare.com/articles/dataset/Taxonomic_vs_genomic_fungi/22692505 (https://doi.org/10.6084/m9.figshare.22692505.v1).

### Code availability
Scripts used for the inference of genome-wide duplication and loss history are available at https://github.com/zsmerenyi/compaRe (ref. [89]).

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

## Acknowledgements

We appreciate the critical comments of B. Papp and E. Ocaña-Pallarès on the earlier version of the manuscript. A. Csikász-Nagy and A. Hubai are thanked for the useful discussions on PCL cyclins and multivariate methods, respectively. We thank M. C. Aime for permission to utilize unpublished genomic data of *Tritirachium* sp. This work was funded by the Momentum Program of the Hungarian Academy of Sciences (LP2019-13/2019) and by the European Research Council (Grant No. 758161) (both to L.G.N.).

## Author contributions

Z.M., L.G.N., K.K, J.S. and J.W.S. designed the research. K.K., N.S., X.-B.L., B.B. and Z.M. performed the data collection and curation. Z.M. performed the comparative analyses and bioinformatics. L.G.N., J.W.S. and Z.M. interpreted the results and wrote the paper. All authors have read and agreed to the published version of the paper.

## Competing interests

The authors declare no competing interests.

## Additional information

**Extended data** is available for this paper at https://doi.org/10.1038/s41559-023-02095-9.

**Correspondence and requests for materials** should be addressed to László G. Nagy.

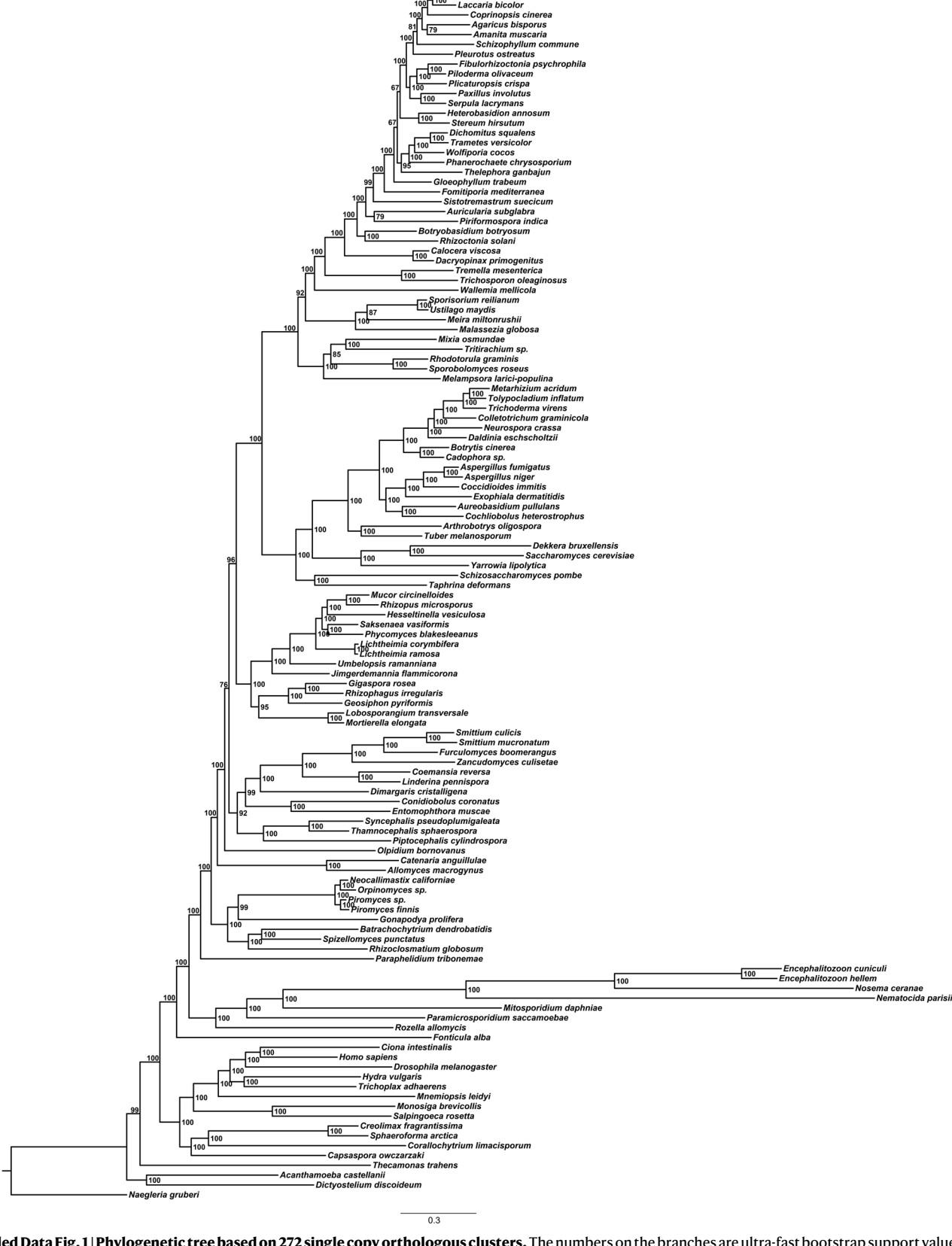

**Extended Data Fig. 1 | Phylogenetic tree based on 272 single copy orthologous clusters.** The numbers on the branches are ultra-fast bootstrap support values inferred under the LG+ G model (ML analysis). The scale bar represents 0.3 expected change per site.

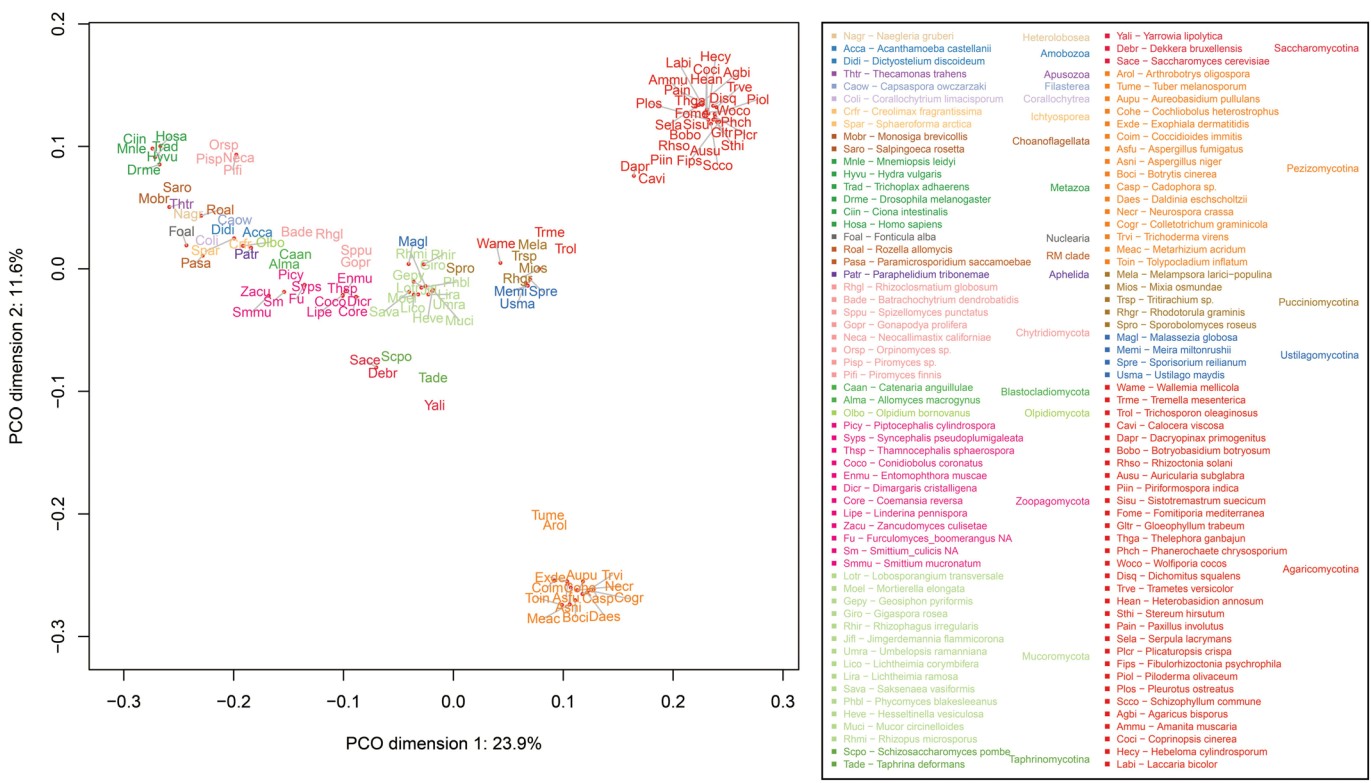

**Extended Data Fig. 2 | Principal coordinate analysis based on homologous groups.** Principal coordinate analysis (PCoA) based on the presence/absence data of 9,993 HGs. Microsporidia species were removed from this analysis for better visualisation.

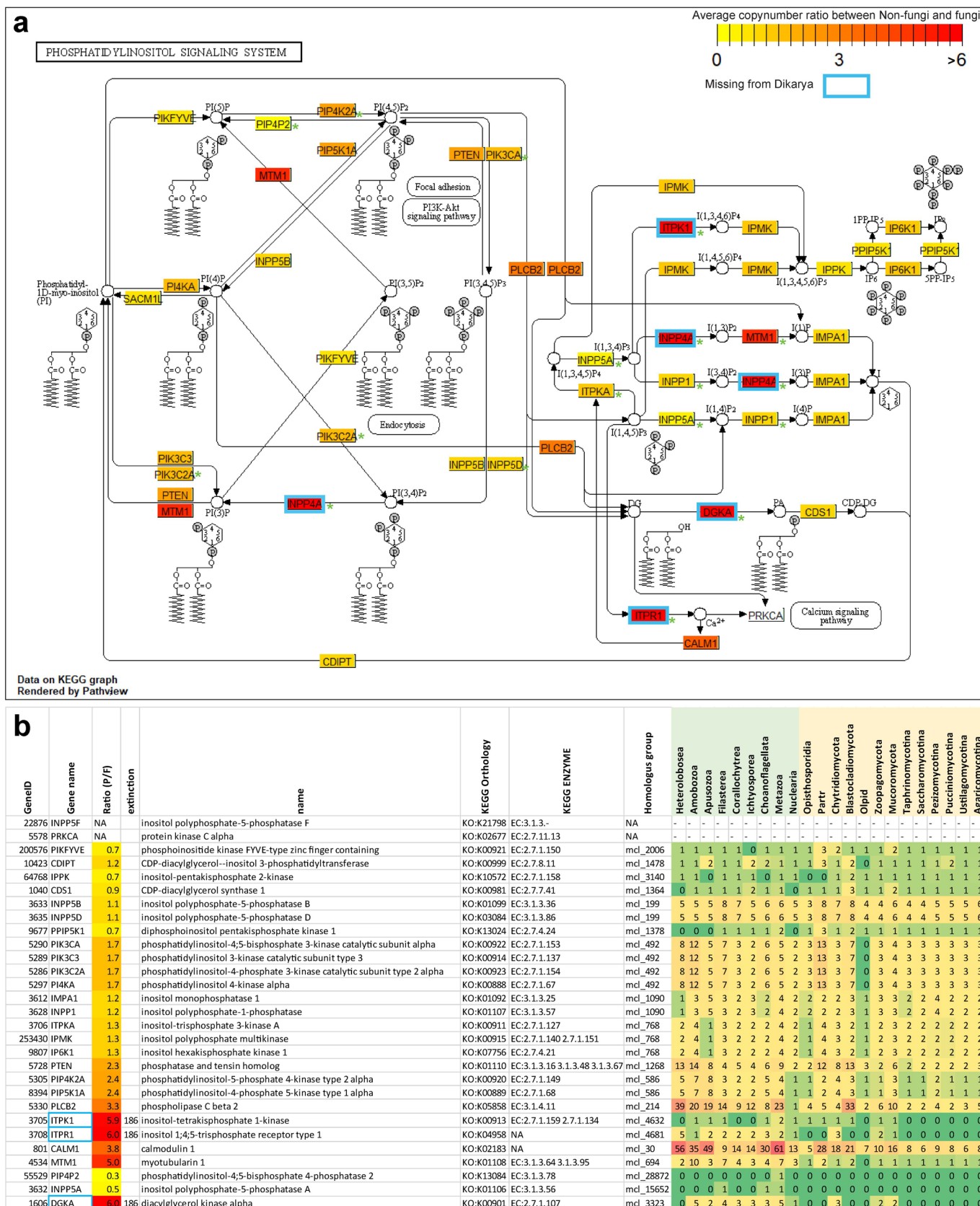

**Extended Data Fig. 3 | See next page for caption.**

**Extended Data Fig. 3 | Decreased redundancy and diversity of the phosphatidylinositol signalling system in fungi. a)** KEGG graph based on *Homos sapiens* gene IDs. A rectangle can contain multiple genes from the same homologous group, therefore we have only shown the representative gene name that KEGG uses, in the chart and table (b) as well. Colouring of the rectangles is based on the average copy number ratio between non-fungi and Fungi (Ratio N/F), warmer colour represents more members of a given HG in non-fungi than in Fungi. Green asterisk (*) indicates the absence of a component in the *Saccharomyces* pathway based on the KEGG pathway (sce04070), while blue stroke represents the absence of a component from Dikarya based on our clustering. **b)** The table shows the average copy numbers of the HGs in which the components of the pathway were clustered. The ratio (N/F) explained above was used to colour the rectangles in diagram (a).

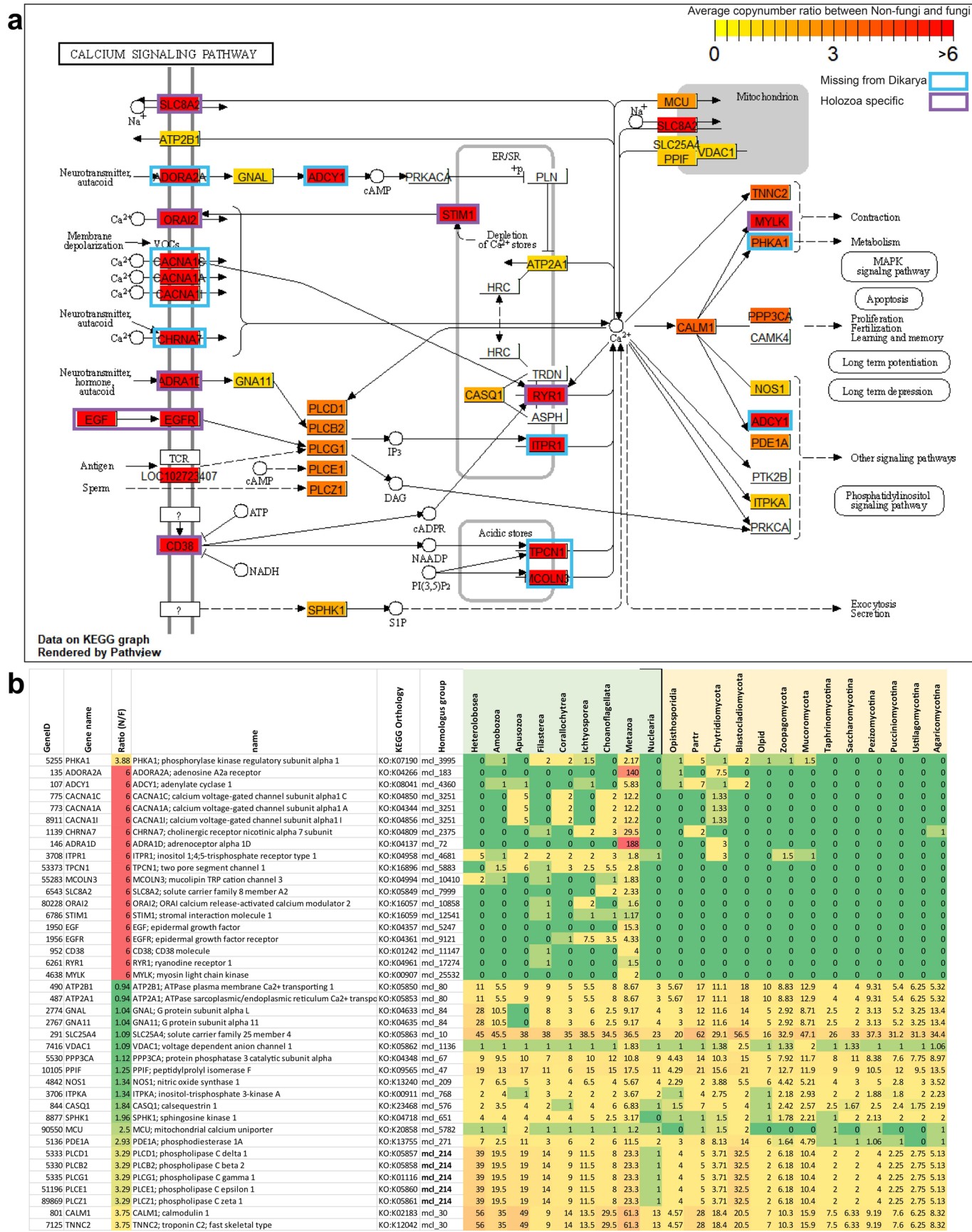

**Extended Data Fig. 4 | See next page for caption.**

**Extended Data Fig. 4 | Decreased redundancy and diversity of the calcium signalling system in fungi. a)** KEGG graph based on *Homos sapiens* gene IDs. A rectangle can contain multiple genes from the same homologous group, therefore we have only shown the representative gene name that KEGG uses, in the chart and table (b) as well. Colouring of the rectangles is based on the average copy number ratio between non-fungi and Fungi (Ratio N/F), warmer colour represents more members of a given HG in non-fungi than in Fungi. Blue stroke represents the absence of a component from Dikarya while purple stroke represents Holozoa specific genes based on our clustering. **b)** The table shows the average copy numbers of the HGs in which the components of the pathway were clustered. The ratio (N/F) explained above was used to colour the rectangles in chart (a).

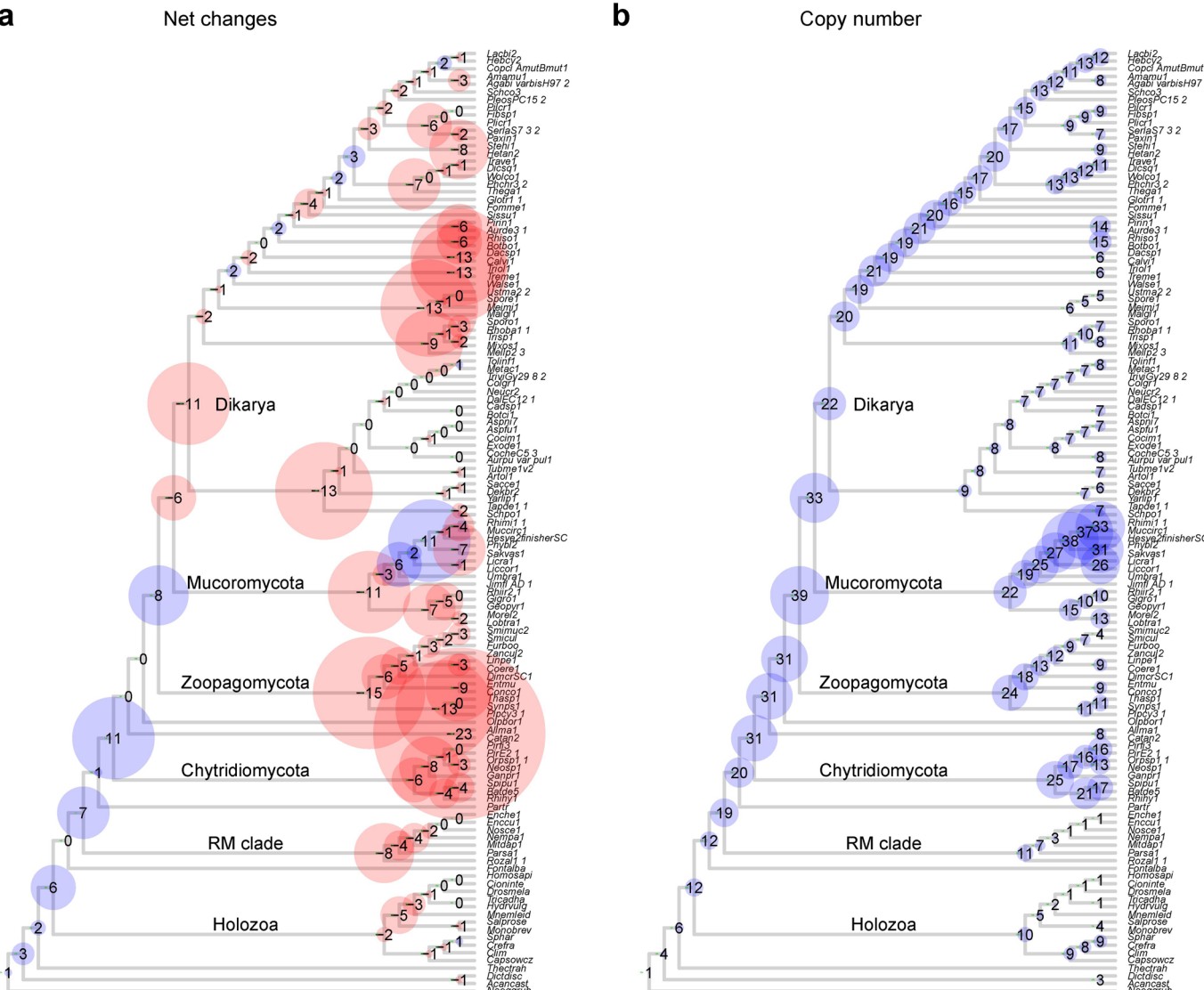

**Extended Data Fig. 5 | Copy numbers and evolution of the PCL cyclin family. a)** Net changes (expansion - blue; contraction - red) to ancestral protein coding capacity across the fungal phylogeny as inferred by Dollo mapping of duplications and losses. **b)** inferred ancestral copy numbers. Duplications mapping to terminals (that is, resulting in species specific paralogs) are not shown. The size of the circles is proportional to the number of net gain events and copy numbers.

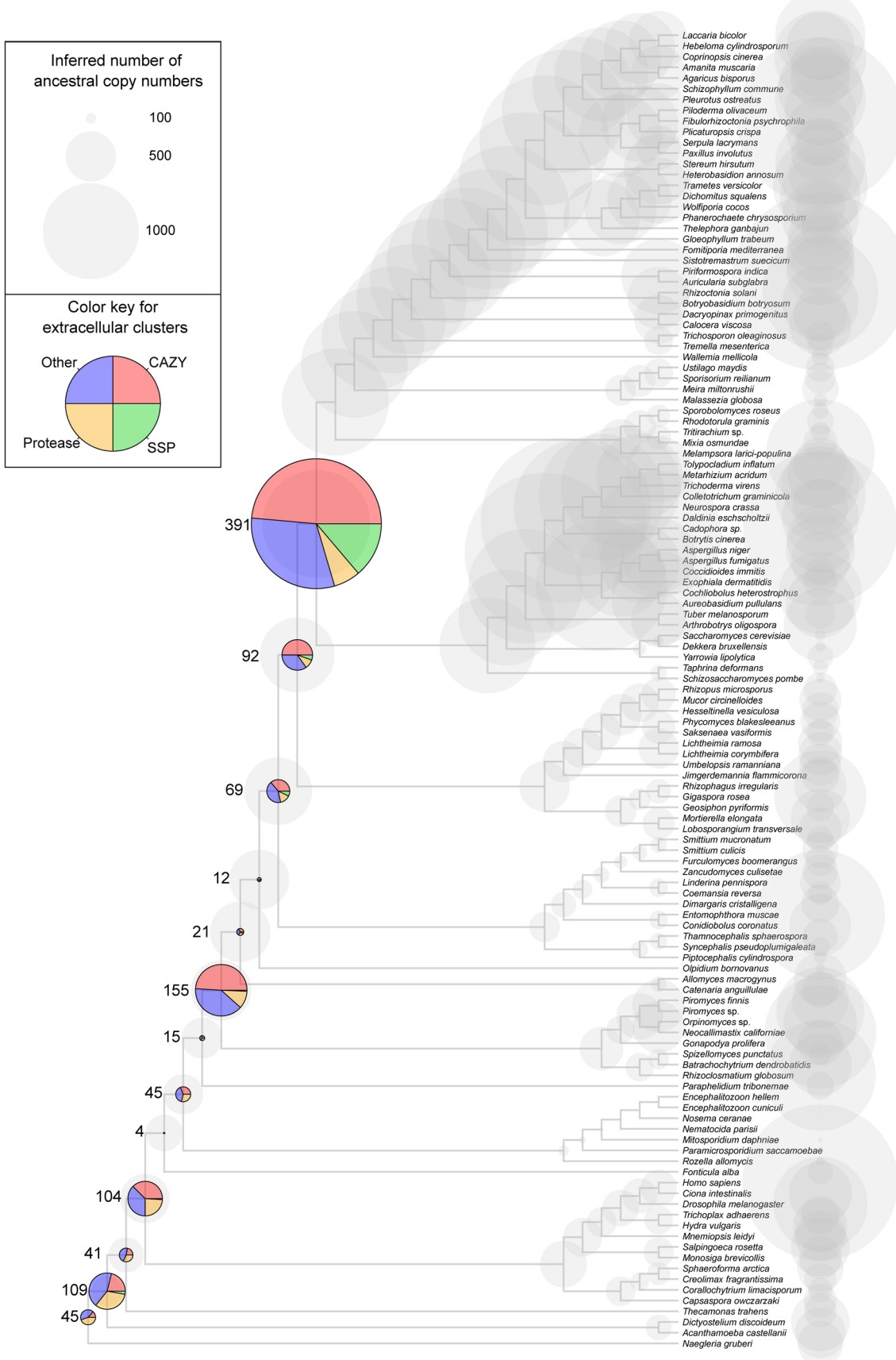

**Extended Data Fig. 6 | Copy numbers and evolution of homologous groups containing extracellular proteins.** Copy numbers are shown as grey circles, while numbers of duplications are represented by pie-charts. Numbers next to pie charts means the number of duplications. The size of the circles is proportional to the number of net gain events and copy numbers.

**a**   Net changes                          **b**   Copy number

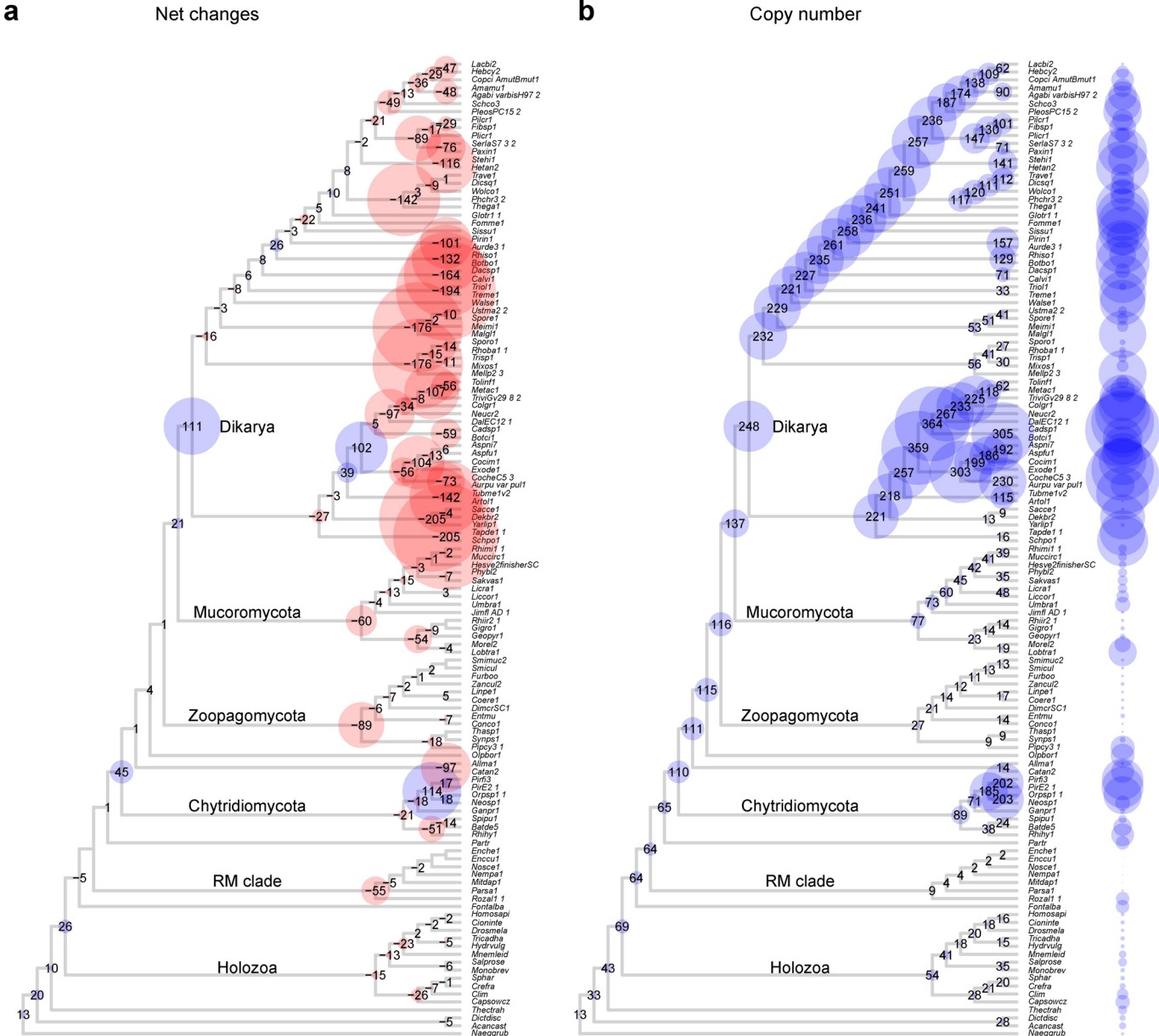

**Extended Data Fig. 7 | Ancestral copy numbers and evolution of plant cell wall degrading enzyme (PCWDE) HGs. a)** Net changes (expansion - blue; contraction - red) and **b)** inferred ancestral protein coding capacity across the fungal phylogeny. Duplications mapping to terminals (that is species specific paralogs) are not shown. The size of the circles is proportional to the number of net gain events and copy numbers in panel (a) and (b), respectively.

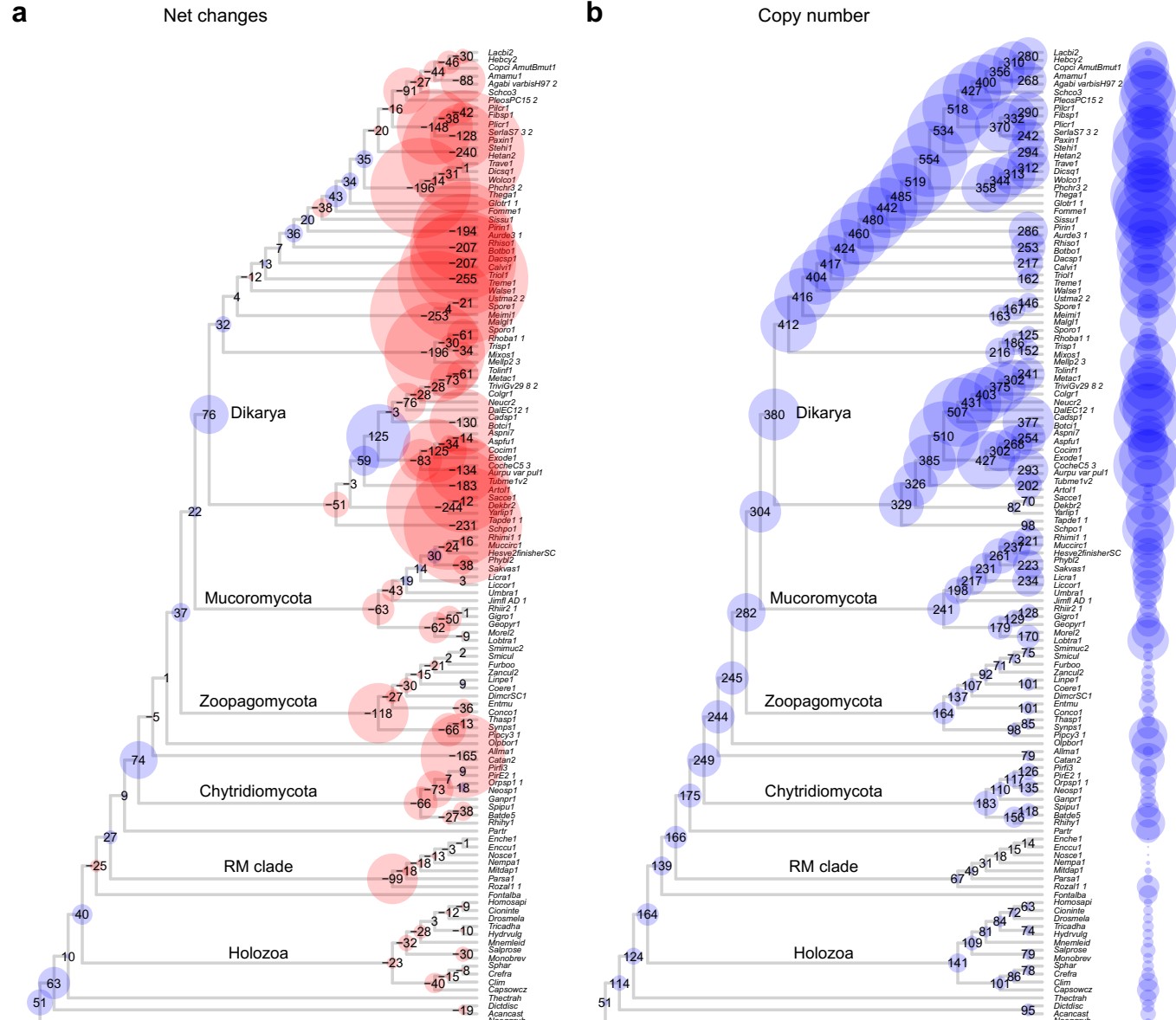

**Extended Data Fig. 8 | Ancestral copy numbers and evolution of fungal cell wall (FCW) and related cell surface proteins. a)** Net changes (expansion - blue; contraction - red) and **b)** inferred ancestral protein coding capacity across the fungal phylogeny. Duplications mapping to terminals (that is species specific paralogs) are not shown. The size of the circles is proportional to the number of net gain events and copy numbers in panel (a) and (b), respectively.

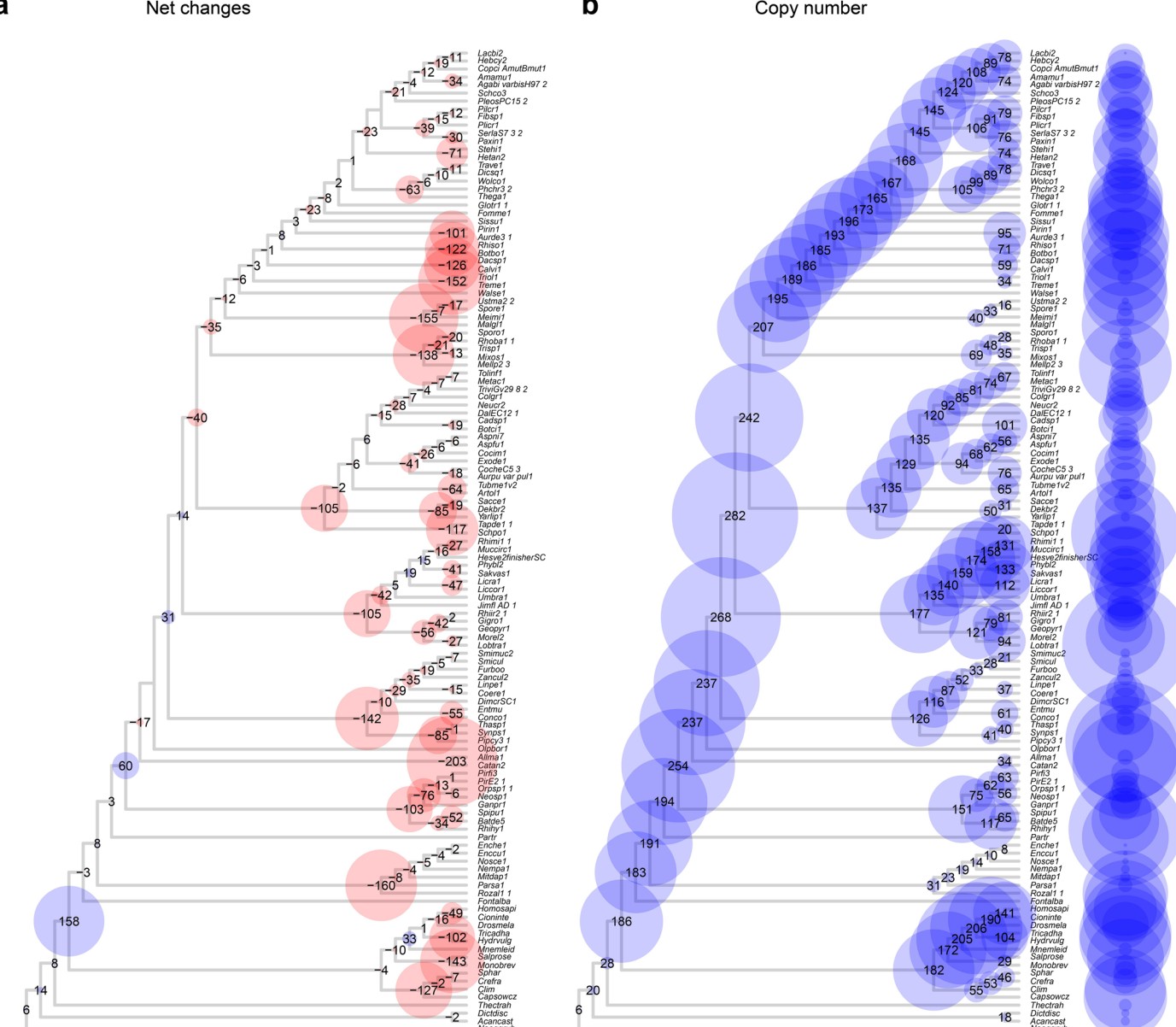

**a** Net changes

**b** Copy number

**Extended Data Fig. 9 | Ancestral copy numbers and evolution of transporters.**
**a)** Net changes (expansion - blue; contraction - red) to ancestral protein coding
capacity across the fungal phylogeny. **b)** inferred ancestral copy numbers.

Duplications mapping to terminals (that is species specific paralogs) are not
shown. The size of the circles is proportional to the number of net gain events and
copy numbers in panel (a) and (b), respectively.

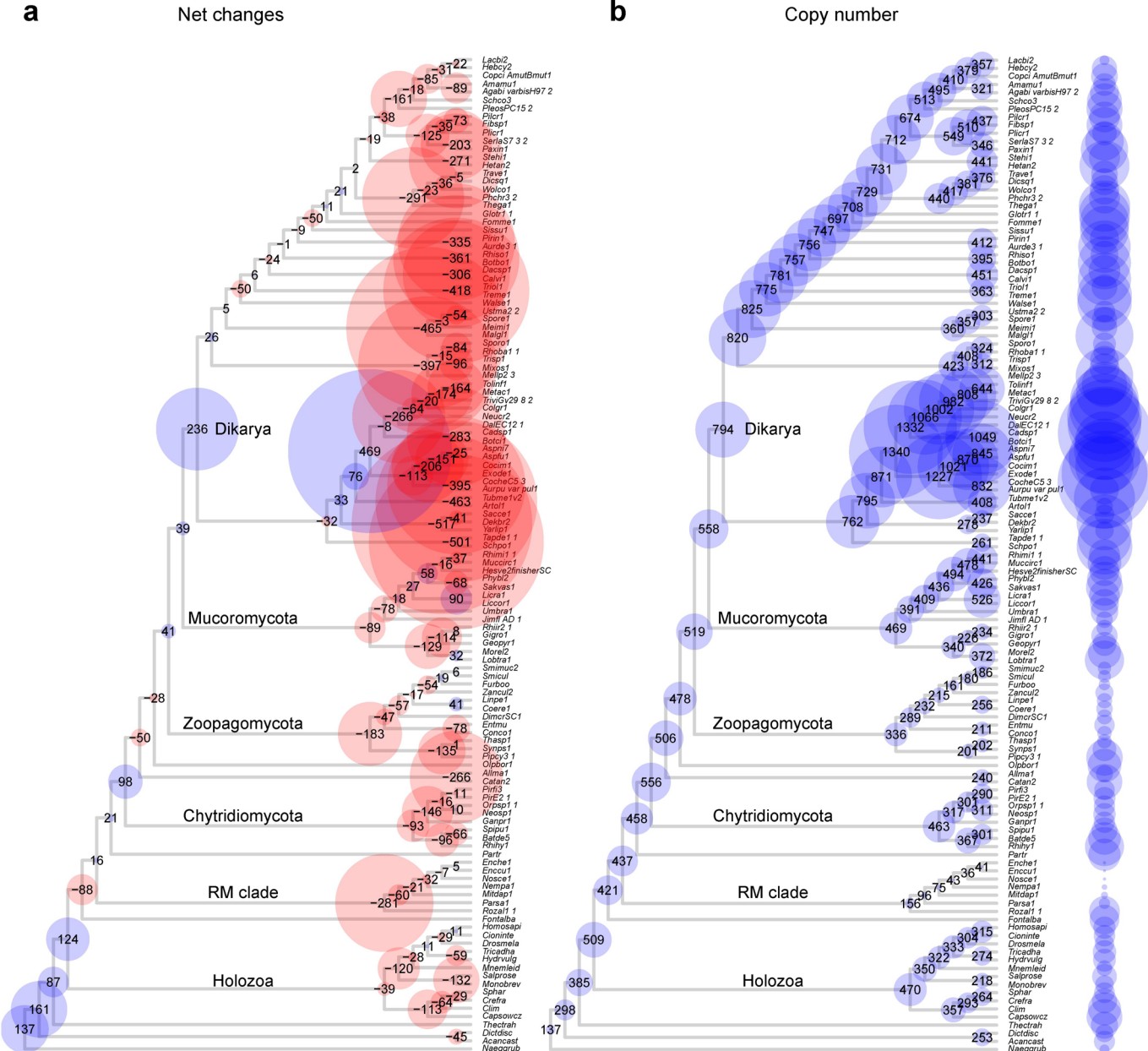

**Extended Data Fig. 10 | Ancestral copy numbers and evolution of the C2H2 transcription factor family. a)** Net changes (expansion - blue; contraction - red) to ancestral protein coding capacity across the fungal phylogeny. **b)** inferred ancestral copy numbers. Duplications mapping to terminals (that is species specific paralogs) are not shown. The size of the circles is proportional to the number of net gain events and copy numbers in panel (a) and (b), respectively.

# Reporting Summary

## Statistics

For all statistical analyses, confirm that the following items are present in the figure legend, table legend, main text, or Methods section.

| n/a | Confirmed | |
|---|---|---|
| ☐ | ☒ | The exact sample size (*n*) for each experimental group/condition, given as a discrete number and unit of measurement |
| ☒ | ☐ | A statement on whether measurements were taken from distinct samples or whether the same sample was measured repeatedly |
| ☐ | ☒ | The statistical test(s) used AND whether they are one- or two-sided *Only common tests should be described solely by name; describe more complex techniques in the Methods section.* |
| ☒ | ☐ | A description of all covariates tested |
| ☒ | ☐ | A description of any assumptions or corrections, such as tests of normality and adjustment for multiple comparisons |
| ☐ | ☒ | A full description of the statistical parameters including central tendency (e.g. means) or other basic estimates (e.g. regression coefficient) AND variation (e.g. standard deviation) or associated estimates of uncertainty (e.g. confidence intervals) |
| ☐ | ☒ | For null hypothesis testing, the test statistic (e.g. *F*, *t*, *r*) with confidence intervals, effect sizes, degrees of freedom and *P* value noted *Give P values as exact values whenever suitable.* |
| ☒ | ☐ | For Bayesian analysis, information on the choice of priors and Markov chain Monte Carlo settings |
| ☒ | ☐ | For hierarchical and complex designs, identification of the appropriate level for tests and full reporting of outcomes |
| ☒ | ☐ | Estimates of effect sizes (e.g. Cohen's *d*, Pearson's *r*), indicating how they were calculated |

*Our web collection on statistics for biologists contains articles on many of the points above.*

## Software and code

Policy information about availability of computer code

| Data collection | InterPro Scan (v.5.46-81.0), MAFFT, TrimAL, RAxML (v8.2.12), FigTree, IQ-Tree v2.0.3, MMseqs2 v13.45111, CAZY database, JGI Mycocosm |
|---|---|
| Data analysis | COMPARE pipeline (available at https://github.com/zsmerenyi/compaRe), R packages used for analysis (topGO, phytools, ape, tidyr, phangorn, stat and tidyverse) |

For manuscripts utilizing custom algorithms or software that are central to the research but not yet described in published literature, software must be made available to editors and reviewers. We strongly encourage code deposition in a community repository (e.g. GitHub). See the Nature Portfolio guidelines for submitting code & software for further information.

## Data

Policy information about availability of data

All manuscripts must include a data availability statement. This statement should provide the following information, where applicable:
- Accession codes, unique identifiers, or web links for publicly available datasets
- A description of any restrictions on data availability
- For clinical datasets or third party data, please ensure that the statement adheres to our policy

Accession codes, unique identifiers, or web links for publicly available datasets

# Human research participants

Policy information about <u>studies involving human research participants and Sex and Gender in Research.</u>

| | |
|---|---|
| Reporting on sex and gender | *Use the terms sex (biological attribute) and gender (shaped by social and cultural circumstances) carefully in order to avoid confusing both terms. Indicate if findings apply to only one sex or gender; describe whether sex and gender were considered in study design whether sex and/or gender was determined based on self-reporting or assigned and methods used. Provide in the source data disaggregated sex and gender data where this information has been collected, and consent has been obtained for sharing of individual-level data; provide overall numbers in this Reporting Summary. Please state if this information has not been collected. Report sex- and gender-based analyses where performed, justify reasons for lack of sex- and gender-based analysis.* |
| Population characteristics | *Describe the covariate-relevant population characteristics of the human research participants (e.g. age, genotypic information, past and current diagnosis and treatment categories). If you filled out the behavioural & social sciences study design questions and have nothing to add here, write "See above."* |
| Recruitment | *Describe how participants were recruited. Outline any potential self-selection bias or other biases that may be present and how these are likely to impact results.* |
| Ethics oversight | *Identify the organization(s) that approved the study protocol.* |

Note that full information on the approval of the study protocol must also be provided in the manuscript.

# Field-specific reporting

Please select the one below that is the best fit for your research. If you are not sure, read the appropriate sections before making your selection.

☐ Life sciences    ☐ Behavioural & social sciences    ☒ Ecological, evolutionary & environmental sciences

For a reference copy of the document with all sections, see nature.com/documents/nr-reporting-summary-flat.pdf

# Ecological, evolutionary & environmental sciences study design

All studies must disclose on these points even when the disclosure is negative.

| | |
|---|---|
| Study description | In this paper we focused on the phylogenomic comparisons of Holomycota genomes with a diverse Amorphean dataset. |
| Research sample | published whole genomes of 123 species |
| Sampling strategy | Sampling of species based on taxonomic classification. We intend to sample all known (until 2022) phyla of the Holomycota clade. |
| Data collection | JGI mycocosm database |
| Timing and spatial scale | These are previously published genomes, therefore no data collection date is application. Versions of each species used in this study are mentioned in Supplementary Data 1. |
| Data exclusions | not applicable for this study |
| Reproducibility | To ensure reproducibility, the codes used in the study are made publicly available on github profile, mentioned in "code availability section of the manuscript. |
| Randomization | N/A |
| Blinding | N/A |

Did the study involve field work?    ☐ Yes    ☒ No

# Reporting for specific materials, systems and methods

We require information from authors about some types of materials, experimental systems and methods used in many studies. Here, indicate whether each material, system or method listed is relevant to your study. If you are not sure if a list item applies to your research, read the appropriate section before selecting a response.

## Materials & experimental systems

| n/a | Involved in the study |
|-----|----------------------|
| ☒ ☐ | Antibodies |
| ☒ ☐ | Eukaryotic cell lines |
| ☒ ☐ | Palaeontology and archaeology |
| ☒ ☐ | Animals and other organisms |
| ☒ ☐ | Clinical data |
| ☒ ☐ | Dual use research of concern |

## Methods

| n/a | Involved in the study |
|-----|----------------------|
| ☒ ☐ | ChIP-seq |
| ☒ ☐ | Flow cytometry |
| ☒ ☐ | MRI-based neuroimaging |

