## [Peer Review File · Nature Ecology & Evolution]

Peer Review Information

Journal: Nature Ecology & Evolution

Manuscript Title: Contrasting patterns of gene gain and loss in early fungal evolution reveals a difference between taxonomic and genomic fungi

Corresponding author name(s): László G. Nagy

Editorial Notes:

Reviewer Comments & Decisions:

Decision Letter, initial version:

7th February 2023

Dear László,

Your manuscript entitled "Taxonomic vs genomic fungi: contrasting evolutionary loss of protistan genomic heritage and punctuated emergence of fungal novelties" has now been seen by three reviewers, whose comments are attached. The reviewers have raised a number of concerns which will need to be addressed before we can offer publication in Nature Ecology & Evolution. We will therefore need to see your responses to the criticisms raised and to some editorial concerns, along with a revised manuscript, before we can reach a final decision regarding publication.

We therefore invite you to revise your manuscript taking into account all reviewer and editor comments. Please highlight all changes in the manuscript text file in Microsoft Word format.

* If you have not done so already please begin to revise your manuscript so that it conforms to our Article format instructions at <http://www.nature.com/natecolevol/info/final-submission>. Refer also to any guidelines provided in this letter.

2[REDACTED]

Nature Ecology & Evolution is committed to improving transparency in authorship. As part of our efforts in this direction, we are now requesting that all authors identified as 'corresponding author' on published papers create and link their Open Researcher and Contributor Identifier (ORCID) with their account on the Manuscript Tracking System (MTS), prior to acceptance. ORCID helps the scientific community achieve unambiguous attribution of all scholarly contributions. You can create and link your ORCID from the home page of the MTS by clicking on 'Modify my Springer Nature account'. For more information please visit <http://www.springernature.com/orcid>.

[REDACTED]

Reviewer expertise:

Reviewer #1: evolutionary genomics, phylogenomics, including protists and fungi

Reviewer #2: evolution of Fungi, phylogenetics

Reviewer #3: evolution of fungi, phylogenomics, comparative genomics

Reviewers' comments:

Reviewer #1 (Remarks to the Author):

In this manuscript the authors tackle a largely discussed and complex topic: Where do we draw the line of fungi and their protists (unicellular eukaryotic) ancestors? One way to do this is to understand the genomic transitions from the unicellular ancestor to the derived multicellular fungi. Thus, the authors try to tackle this problem by using comparative genomics to study 123 whole genomes and

2analyze the temporal and functional trends in genomic changes. The authors recovered that early diverging fungi are genetically intermediate between pre-fungal protists and Dikarya. They also observed that in Fungi, the shedding of ancient protist gene families happened in parallel with the emergence of fungal novelties and waves of expansion of preexisting families. Their most important result it's a big one, and its showing that taxonomically defined fungi don't match with 'genomic fungi': 'Protists' is a continuum of organisms that extends deep within what it has traditionally being defined as fungi. The paper is carefully drafted, the methods (for which I have no comments) and thus, its results, are solid, making it an essential read and contribution for the community. The implications of this study extend beyond the fields of mycology and protistology. I consider the manuscript is almost ready to publish as it is, I only have one major comment and a couple of minor comments that need to be addressed.

The main comment I have is that I consider important that the authors discuss and address osmotrophy as a possible clade(fungi?)-defining trait. The largest gene gain events the authors recover (C and D) are related to GO terms involved in the transition towards osmotrophy (lines 204 to 209), right at the base of Chytridiomycota+all other fungi and at the base of Dikarya, (this does not happen at the base of aphelids+fungi). Additionally, they found that for highly duplicated gene groups, osmotrophy related genes were the predominant class (extracellular proteins, transporters, etc.) (lines 359 to 362). Even when osmotrophy evolved independently in microsporidia, it is clear that it does not account for the same amount of genomic changes as the one that underwent at node C. Osmotrophy on microsporidia appeared as a product of genome-reduction and HGT of transporters. However, (as the authors show) osmotrophy in Chytridiomycota and all other fungi (D) is a product of gene gain and duplication. Osmotrophy in the clade of Chytrids+all other fungi and osmotrophy in microsporidia are two different things. Thus, for example, referring to chytrids and aphelids (clade Phytophagea) as 'Early diverging fungi' (line 82 to 83) may or may not be the most adequate due to one clade being phagotrophic and the other one osmotrophic. As shown by the authors the Chytridiomycota split is an important one (line 141), thus defining the group of Chytridio+Blastocladio+Sanchytrio+Olpidio+Zoopago+etc. as a taxonomic entity may still be possible. I consider the discussion around osmotrophy very important since it directly derives from the authors results, and one that will help the community to better define these clades.

Other comments:

Lines 46 to 48: The authors need to introduce Holomycota in the introduction, and I think that a good moment to do it is on lines 46 to 48, where they include as 'Kingdom Fungi' almost all members of Holomycota.

Line 37 and Line 46 and other places of the text: The word Kingdom is problematic (since it mostly implies the outdated Whittaker classification) and beyond the discussion on where to draw the line of Fungi, I would recommend eliminating the word kingdom in all of the manuscript and referring to Fungi (Chytridiomycota + Blastocladiomycota + Sanchytriomycota + Zoopagomycota + Mucoromycota + Glomeromycota + Dikarya) + Aphelida + Rozellida + Microsporidia + nucleariids as Holomycota or Nucleomycea, whenever is possible. For example, when using the term 'kingdom-level clades' in line 37, I would suggest instead: supergroup-level, high-rank level, etc. I do understand Kingdom is an easy way to refer the clade, the same way the word 'protist' is to talk about unicellular eukaryotes. However, I think we need to make an effort to eliminate inaccurate terms little by little, especially in a

3high-quality and high-impact study as this one.

Line 47 and line 346: Opisthosporidia is a clade created for Rozellida(Rozellomycota)+Microsporidia+Aphelida, not only for Rozellida+Microsporidia. Since it has been extensively proven that aphelids are sister to fungi, Opisthosporidia is then a paraphyletic clade and should not be used anymore (at least no without caution). I also recommend referring to the clade of Microsporiadia+Rozellida as Opisthophagea and to Fungi+Aphelids as Phytophagea, as recently suggested by Galindo et al. (10.1093/sysbio/syac054).

Reviewer #2 (Remarks to the Author):

This paper will fascinate biologists interested in broad patterns of evolution across kingdoms. Most convincing and interesting is the analysis of 'Gene groups with the largest contribution to genomic change'. I enjoyed Figs. 3 & 4, and the supplements showing patterns of detection of homologs across lineages. However, to fulfill its potential, the authors could dig more deeply into analysis and interpretation, and take more care with language.

Specific criticism.

Language

Starting at the beginning, the title and abstract sounds good but are ultimately confusing. The title suggests that evolution is punctuated or in abstract, highly episodic but rates of change are not analyzed. How is 'episodic' to be reconciled with 'Constant and gradual' gene loss (text in abstract)?

In the title and throughout, meaning of 'protist' is unclear and truly confusing. Protists are the most diverse of eukaryotes and only a small fraction of them are relevant to this paper. Edit to avoid using 'protist' and specify the lineages that are actually being investigated. I only realized what the manuscript meant by 'protist' on looking at the taxa included in the supplement. Formal taxonomy is avoided in the manuscript but loss of meaning resulted.

Taxonomic vs genomic fungi as in title, abstract and conclusions? For some years, in thoughtful publications, different people have chosen different, arbitrary definitions of fungi, all perfectly consistent with phylogeny and phylogenomics. Where do authors want to go with this? What can be contributed that is new?

Edit throughout to take a broader view of fungal evolution. 'archetypal fungal genome' (abstract)? Is one fungal clade's genome more archetypal than others? Animal phylogeneticists avoid the human as archetype; plant phylogeneticists avoid Arabidopsis as archetype.

'Backbone' is used in the manuscript to mean the succession of ancestors to the mushroom lineage. But each ancestral node gives rise to two clades, not one 'backbone'. 'Backbone' could equally well mean a series of ancestors of Chytridiomycota or Mucormycota.

4'two large duplication events' Why two? Fig. 3 shows more inferred high numbers of gains of homology groups along various branches.

In 49. If Mucoromycota are early diverging, their sister, Dikarya must also be early diverging.

Minimize abbreviations for accessibility. For example, rather than LUFA and EDF, taxon names or a short phrase would be easier for readers.

p. 2 In 72. 'Early diverging fungi are intermediate between protists and the Dikarya' What does this mean? Within and among all phyla are large numbers of gains and losses of detectable homology groups.

Analysis

What does 'gradual turnover mean and what is the evidence for it, considering the overall pattern of changes mapped to branches?

103 Gradual turnover of protist heritage and synapomorphies in fungi

This relates to Figure 2. How do the numbers in Fig. 2 relate to the much larger numbers of losses and gains of homology groups shown in Fig. 3? What's a fungal novelty? If protists are polyphyletic, how can genes be protist specific?

To provide necessary context for the proportion of gains and losses of homology groups, give the total number of inferred homology groups in addition to the number of changes across the tree, not just in the lineages ancestral to Basidiomycota in Figs 3 and 4. Giving the proportion of transcription factors and other kinds of genes that change will better show how broad the rewiring of transcriptional networks might have been.

Every branch leading to a phylum has at least twice as many inferred losses of homology groups relative to gains while their sister branches do not. This is odd. Is it an artifact of the methods, perhaps reflecting an inability to track homology groups through longer branches that reflect longer periods of time vs shorter branches among phyla that represent shorter time intervals? or is there some underlying evolutionary explanation? What's happening to the homologs? Is there domain shuffling that interrupts the query coverage? Check analytically; is there a correlation between inferred branch length and inferred proportion of loss of homology groups?

If there is a correlation between length of branch and inferred proportion of changes in homology groups, that points to some constancy of evolution, not episodic evolution. If the proportion of changes over time varies greatly, that would support the idea of punctuated change. In a couple places, the manuscript discusses rates when it only shows numbers of changes. Either rates should be analyzed, or 'rates' should be replaced by 'number of changes' or whatever else is appropriate.

Figure 1. It would be more interesting to see retention of homologs mapped onto a phylogeny than used in a PCA because for example, the proximity of smuts, yeasts and zygomycetes is probably due to convergent losses not to their intermediate evolutionary position. Proximity can't be interpreted as indicating evolutionary intermediates.

Figures and conclusions

Figure 1. The PCA should be uncluttered by using letters, not circles to more clearly designate clade membership. Even zooming in to look at the high resolution download, it's hard to see which data points are from which clades. Remove the icons because they're available just to the right, and removing the dotted lines. The shading in the polygons is too bright and again, it makes the points hard to see.

Conclusions will benefit from being tightened up. At present, they do not highlight the best supported aspects of the paper but are a collection of inadequately supported statements.

Figure 3 shows the same pattern of change in holozoa and metazoa as in fungal phyla. What's the basis for arguing for a contrast in patterns in different kingdoms? 327 content has changed drastically during early fungal evolution. However, in contrast to animals³ and 328 plants², this has not happened abruptly at the taxonomic limit of the kingdom, even if competing 329 taxonomic circumscriptions of Fungi are considered.

Line 355 Wording needs re-consideration 'in which gene loss aided the gradual shedding of protist traits ' Exactly which genes and traits are meant here and what evidence suggests that gene loss helped lose ancestral traits? Doesn't gene loss usually follow loss of selection for function, as with the flagellar apparatus, where gene loss seems to be quite quick after flagellar apparatus (an ancestral structure) was lost from various lineages. Opisthosporidia underwent drastic genome reduction, not gradual gene loss.

Line 378. Not much evidence of 'a unicellular algal parasite ancestor'. Nucleariids seem able to digest bacteria and assorted microbes and Opisthosporidia are endoparasites. With the diversity among extant taxa that diverged from early within fungi, it's unreasonable to assume that the ancestral fungus was aphelid-like.

Fig. S3. Explain P/F

Reviewer #3 (Remarks to the Author):

6Merenyi et al have profiled the genomic content of 123 fungi and related eukaryotes to attempt to chart evolutionary trends within the fungal kingdom. The authors find that the evolution of Fungi arising from Opisthokonts and their subsequent genomic development was episodic in nature, marked by gradual turnover of "protistan" gene content across all major fungal lineages and more interspersed events of probable gene family and/or genome duplication. This turnover appears to be more pronounced among genes related to transport and transcription processes. Compared to other eukaryote groups of similar rank, such as Metazoa, Fungi appear to lack distinct synapomorphies that would be commonly used to distinguish other major eukaryote nodes. Taken as a whole, the authors suggest that there is some dissonance between how we classify "Fungi" taxonomically and their relative lack of synapomorphies at the genomic level. This seems a reasonable conclusion given the fact that this type of analysis encompasses millions of years of different evolutionary processes and fungi have evolved very differently from plants or animals.

I think this manuscript is of appreciable quality to anyone studying fungal or eukaryote genome evolution and would expect to see this kind of work in a journal like NEE. The approach and methodology outlined in this study would also be suited to other tricky parts of the eukaryote tree. I have some suggestions below which hopefully will improve upon the manuscript in its current state, but otherwise I would recommend this for publication pending some revisions.

Comments:

Overall there are several typographical errors in this manuscript such commas being in places they don't need to be in, and some sentences that could be rephrased or broken up to improve readability. I think this would not require much work outside of simply reading over the manuscript once or twice. Otherwise, the figures and supplementary information provided are clear and easy to understand.

Introduction

Pg 1, Lines 38: Not sure the "conversely" is needed here, this is referring to two notions (identifying the genomic divergence of a new lineage from its relative and the subsequent innovations within that lineage) that aren't necessarily contrasting in my view.

Pg 2, Lines 46-61: It may be worth rephrasing this at the start to contrast the general division of Fungi into the (generally) sessile osmotrophs of the Dikarya, which is the part of "Fungi" most readers will probably be familiar with, and the EDF which are harder to distinguish from protists and would need outlining. This can then lead into the discussion of LUFAs. At the moment, the mention of Dikarya comes at the end.

Pg 2, Lines 59-60: "However" appears in this sentence twice.

Results

Pg 2, Line 75: Unnecessary commas such as "all, but one". Is Sanchythriomycota a typo?

Pg 3, Figure 1b: If possible, I'd appreciate if the authors could include the full PCoA plot with species labelling in the supplementary material like they have with the phylogenetic tree. For me this is the most striking figure in the manuscript, and it would be interesting to see internal relationships within those phylum clusters. For example, I'd be curious to know if the broad-looking distribution of the

7chytrids in the PCoA plot is related to the divide between the Neocallimastigo- group and other chytrids.

Pg 3, Line 104-111: This is presented in a slightly confusing manner. There are 540 HGs that are absent in Dikarya but are present in other all sampled Holomycota (EDF+Nucleariids in this case). LUFA lost 41 of these when it diverged from its closest sampled relatives, the Nucleariids. I would rephrase this paragraph. In Figure 2, this number is given as 31.

Pg 3, Line 119: The authors mention later on (Line 131) that GO annotation is more precise in metazoans than in Fungi. One could argue that the relative bias of GO annotations towards human and model organism data creates an artificial limit on what we can say about their distribution in non-model organisms. Arguably, the authors observed this themselves by the number of unannotated HGs they found (Lines 155-157). I think the authors could highlight the limitations of GO enrichment analysis as is currently implemented in this regard, even though it is a perfectly reasonable analysis to perform otherwise.

Pg 4, Line 124-5: Some typos, I would use "despite" instead of "despite of" and "between" instead of "among".

Pg 4, Line 128: The authors use the term "RM clade" here to refer to Rozellomycota+Microsporidia, but in the Introduction and Conclusion they use the term "Opisthosporidia".

Pg 4, Line 142: There's a typo in the citation here, "see 4th chapter"? When the authors refer to the "split of chytrids" as they do later in the manuscript, it's not immediately obvious as to whether they mean the divergence of the chytrids from other Fungi or to internal divergence within the chytrids.

Pg 6, Lines 218-219: My understanding is that the Neocalli- group are almost exclusively comprised of species found in the gut of ruminants. I don't know whether that environmental limitation has anything to do with the lineage-specific duplications seen in Figure 3, or whether those environments are any more conducive to bacteria-fungi HGT than e.g. soil. The manuscript shies away from discussing HGT otherwise, which is fine, but raising the subject here may require some additional explanation.

Conclusion

No comments.

Methods

Pg 10, Line 382: I would include the reference for Sanchytriomycota (I presume this is Galindo et al 2021 in Nature Communications).

Pg 11, Line 434: I would include the word "analysis" after the mention of InterProScan.

*****END*****

Author Rebuttal to Initial comments

Response to Reviewers' comments:

Reviewer #1 (Remarks to the Author):

In this manuscript the authors tackle a largely discussed and complex topic: Where do we draw the line of fungi and their protists (unicellular eukaryotic) ancestors? One way to do this is to understand the genomic transitions from the unicellular ancestor to the derived multicellular fungi. Thus, the authors try to tackle this problem by using comparative genomics to study 123 whole genomes and analyze the temporal and functional trends in genomic changes. The authors recovered that early diverging fungi are genetically intermediate between pre-fungal protists and Dikarya. They also observed that in Fungi, the shedding of ancient protist gene families happened in parallel with the emergence of fungal novelties and waves of expansion of preexisting families. Their most important result it's a big one, and its showing that taxonomically defined fungi don't match with 'genomic fungi': 'Protists' is a continuum of organisms that extends deep within what it has traditionally being defined as fungi. The paper is carefully drafted, the methods (for which I have no comments) and thus, its results, are solid, making it an essential read and contribution for the community. The implications of this study extend beyond the fields of mycology and protistology. I consider the manuscript is almost ready to publish as it is, I only have one mayor comment and a couple of minor comments that need to be addressed.

Answer: We appreciate these words and have revised the ms in accordance with the suggestions given.

The main comment I have is that I consider important that the authors discuss and address osmotrophy as a possible clade(fungi?)-defining trait. The largest gene gain events the authors recover (C and D) are related to GO terms involved in the transition towards osmotrophy (lines 204 to 209), right at the base of Chytridiomycota+all other fungi and at the base of Dikarya, (this does not happen at the base of aphelids+fungi). Additionally, they found that for highly duplicated gene groups, osmotrophy related genes were the predominant class (extracellular proteins, transporters, etc.) (lines 359 to 362). Even when osmotrophy evolved independently in microsporidia, it is clear that it does not account for the same amount of genomic changes as the one that underwent at node C. Osmotrophy on microsporidia appeared as a product of genome-reduction and HGT of transporters. However, (as the authors show) osmotrophy in Chytridiomycota and all other fungi (D) is a product of gene gain and duplication. Osmotrophy in the clade of Chytrids+all other fungi and osmotrophy in microsporidia are

9two different things. Thus, for example, referring to chytrids and aphelids (clade Phytophagea) as 'Early diverging fungi' (line 82 to 83) may or may not be the most adequate due to one clade being phagotrophic and the other one osmotrophic. As shown by the authors the Chytridiomycota split is an important one (line 141), thus defining the group of Chytridio+Blastocladio+Sanchytrio+Olpidio+Zoopago+etc. as a taxonomic entity may still be possible. I consider the discussion around osmotrophy very important since it directly derives from the authors results, and one that will help the community to better define these clades.

Answer: We appreciate this advice. We rephrased the sentence L82-83 (L98-100 in the all mark-up version). Also, we incorporated the suggestion on the Chytridiomycota split in the ms (L414-419). However, because the contradiction between the genome content and the taxonomic borders of fungi would not change by considering other definitions for Fungi, we would not want to make strong suggestions related to taxonomic rearrangements or definitions of fungi. For this reason, we have avoided recommending formal taxonomic changes.

Regarding the suggestion on osmotrophy: Since, as the Reviewer also mentions, osmotrophy evolved independently in Phytophagea and Opisthophagea (and also in the Oomycota), we think it would not be a good trait to define fungi. Therefore, although osmotrophy is undoubtedly a key trait that drove genomic change (and we highlight this better in the revised version L359-367), we think it is not suitable to be considered explicitly as a synapomorphy.

Other comments:

Lines 46 to 48: The authors need to introduce Holomycota in the introduction, and I think that a good moment to do it is on lines 46 to 48, where they include as 'Kingdom Fungi' almost all members of Holomycota.

Answer: We added the description of Holomycota to the introduction.

Line 37 and Line 46 and other places of the text: The word Kingdom is problematic (since it mostly implies the outdated Whittaker classification) and beyond the discussion on where to draw the line of Fungi, I would recommend eliminating the word kingdom in all of the manuscript and referring to Fungi (Chytridiomycota + Blastocladiomycota + Sanchytriomycota + Zoopagomycota + Mucoromycota + Glomeromycota + Dikarya) + Aphelida + Rozellida + Microsporidia + nucleariids as Holomycota or Nucleotmycea,

10

whenever is possible. For example, when using the term 'kingdom-level clades' in line 37, I would suggest instead: supergroup-level, high-rank level, etc. I do understand Kingdom is an easy way to refer the clade, the same way the word 'protist' is to talk about unicellular eukaryotes. However, I think we need to make an effort to eliminate inaccurate terms little by little, especially in a high-quality and high-impact study as this one.

Answer: We appreciate these suggestions, and corrected the text accordingly.

Line 47 and line 346: Opisthosporidia is a clade created for Rozellida(Rozellomycota)+Microsporidia+Aphelida, not only for Rozellida+Microsporidia. Since it has been extensively proven that aphelids are sister to fungi, Opisthosporidia is then a paraphyletic clade and should not be used anymore (at least no without caution). I also recommend referring to the clade of Microsporidia+Rozellida as Opisthophagea and to Fungi+Aphelids as Phytophagea, as recently suggested by Galindo et al. (10.1093/sysbio/syac054).

Answer: We corrected these points.

Reviewer #2 (Remarks to the Author):

This paper will fascinate biologists interested in broad patterns of evolution across kingdoms. Most convincing and interesting is the analysis of 'Gene groups with the largest contribution to genomic change'. I enjoyed Figs. 3 & 4, and the supplements showing patterns of detection of homologs across lineages. However, to fulfill its potential, the authors could dig more deeply into analysis and interpretation, and take more care with language.

Answer: We appreciate the Reviewer's positive comments.

Specific criticism.

Language

Starting at the beginning, the title and abstract sounds good but are ultimately confusing. The title suggests that evolution is punctuated or in abstract, highly episodic but rates of change are not analyzed. How is 'episodic' to be reconciled with 'Constant and gradual' gene loss (text in abstract)?

11Answer: We appreciate pointing this out, and have rephrased the abstract to clarify these points.

In the title and throughout, meaning of 'protist' is unclear and truly confusing. Protists are the most diverse of eukaryotes and only a small fraction of them are relevant to this paper. Edit to avoid using 'protist' and specify the lineages that are actually being investigated. I only realized what the manuscript meant by 'protist' on looking at the taxa included in the supplement. Formal taxonomy is avoided in the manuscript but loss of meaning resulted.

Answer: We have added a clarification on how we define 'protists' in results and discussion (L87-90), methods (L450-451) and in Fig 1a. Also we specified the lineages or changed the word usage in many places.

Taxonomic vs genomic fungi as in title, abstract and conclusions? For some years, in thoughtful publications, different people have chosen different, arbitrary definitions of fungi, all perfectly consistent with phylogeny and phylogenomics. Where do authors want to go with this? What can be contributed that is new?

Answer: We found that the taxonomic border of fungi does not coincide with significant genetic change, and formulated this as 'genomic and taxonomic fungi don't match with each other'. The contradiction between the genomic content and the taxonomic borders of fungi would not change by considering other taxonomic definitions for Fungi, making this a conclusion that we think stands irrespective of which taxonomy one prefers. At the same time, we have intentionally avoided the formulation of a taxonomic standpoint in this ms.

Edit throughout to take a broader view of fungal evolution. 'archetypal fungal genome' (abstract)? Is one fungal clade's genome more archetypal than others? Animal phylogeneticists avoid the human as archetype; plant phylogeneticists avoid Arabidopsis as archetype.

Answer: We wanted to refer to the typical filamentous fungal type genome (compact, with short intergenic regions), but possibly the term archetypal was misleading, so we changed it.

'Backbone' is used in the manuscript to mean the succession of ancestors to the mushroom lineage. But each ancestral node gives rise to two clades, not one 'backbone'. 'Backbone' could equally well mean a series of ancestors of Chytridiomycota or Mucormycota.

Answer: We removed the term from the ms.

'two large duplication events' Why two? Fig. 3 shows more inferred high numbers of gains of homology groups along various branches.

Answer: Rephrased.

In 49. If Mucoromycota are early diverging, their sister, Dikarya must also be early diverging.

Answer: We agree with the reviewer and changed 'early diverging fungi' to "non-Dikarya fungi", which we think is a more appropriate phrase.

Minimize abbreviations for accessibility. For example, rather than LUFAs and EDFs, taxon names or a short phrase would be easier for readers.

Answer: We intend to avoid unnecessary abbreviations, however we think in the case of the acronyms LUFAs, the expanded version would be too long and would also be at the expense of readability in the main text and figures as well. We have removed several abbreviations from the MS (e.g. SSP, EP, EDF).

p. 2 In 72. 'Early diverging fungi are intermediate between protists and the Dikarya' What does this mean? Within and among all phyla are large numbers of gains and losses of detectable homology groups.

Answer: Based purely on gene content-based PCoA analyses we detected a transitional localization of non-Dikarya fungi. Because this analysis ignores evolutionary relationships, we think the term intermediate describes the main result we have here properly. As an alternative outcome, we could have seen a strong separation of holozoan and holomycotan lineages (e.g. similar to the 'branching' seen in the Dikarya), but that is not what we saw. We think this analysis is a good starting point from which we can proceed to unpack the causes.

Analysis

What does 'gradual turnover mean and what is the evidence for it, considering the overall pattern of changes mapped to branches? 103 Gradual turnover of protist heritage and synapomorphies in fungi

Answer: We changed the title of this paragraph to “Gradual loss of protist heritage and emergence of synapomorphies in fungi” (L121).

This relates to Figure 2. How do the numbers in Fig. 2 relate to the much larger numbers of losses and gains of homology groups shown in Fig. 3?

Answer: The numbers in Fig. 2 indicate the number of conserved ancient (red) and fungal-specific (blue) HGs at each node. They represent only the number of HGs, not the number of duplications (of which there might be many within a single HG). In contrast, Fig. 3 indicated all gains (novel HG, gene duplications or even HGT events) and losses derived from orthologous groups.

What's a fungal novelty?

Answer: Rephrased.

If protists are polyphyletic, how can genes be protist specific?

Answer: We appreciate this remark, the term ‘protist-specific’ has been eliminated from the MS.

To provide necessary context for the proportion of gains and losses of homology groups, give the total number of inferred homology groups in addition to the number of changes across the tree, not just in the lineages ancestral to Basidiomycota in Figs 3 and 4. Giving the proportion of transcription factors and other kinds of genes that change will better show how broad the rewiring of transcriptional networks might have been.

Answer: We are not entirely sure what the Reviewer means here. Exact numbers have been added next to all nodes in Fig 3.

Every branch leading to a phylum has at least twice as many inferred losses of

homology groups relative to gains while their sister branches do not. This is odd. Is it an artifact of the methods, perhaps reflecting an inability to track homology groups through longer branches that reflect longer periods of time vs shorter branches among phyla that represent shorter time intervals? or is there some underlying evolutionary explanation? What's happening to the homologs? Is there domain shuffling that interrupts the query coverage?

Answer: We think homology relations are discoverable across both longer and shorter branches. Also to improve the completeness of our clusters (HGs), we applied a cluster-merging method, which is detailed in the method (L466-468). These losses could be explained by the biphasic model of punctuated genome evolution (Wolf & Koonin 2013 Bioessays). According to this hypothesis, between rapid genomic innovations, reduction of genomes is the dominant trend. This has been partially supported by recent publications finding gene loss and genome reduction as the dominant trend in the early evolution of animals (Fernández & Gabaldón 2020, Nat. Ecol. Evol.; Richter et al. 2018 Elife) and plants (Bowles et al 2020 Curr.Biol.).

Check analytically; is there a correlation between inferred branch length and inferred proportion of loss of homology groups? If there is a correlation between length of branch and inferred proportion of changes in homology groups, that points to some constancy of evolution, not episodic evolution. If the proportion of changes over time varies greatly, that would support the idea of punctuated change.

Answer: Thank you for your suggestion on this analysis. We have visualised the total number of gains (left) and losses (right) versus edge length for each node. Neither show any correlation, therefore we think that the HG expansion and contraction events cannot be explained simply by topological reasons or methodological artefacts (e.g. homology detection, protein clustering). According to these results we can reject constancy or time dependent manner of gene repertoire evolution.

In a couple places, the manuscript discusses rates when it only shows numbers of changes. Either rates should be analyzed, or 'rates' should be replaced by 'number of changes' or whatever else is appropriate.

Answer: We deleted the term 'rate' where it was not appropriate.

Figure 1. It would be more interesting to see retention of homologs mapped onto a phylogeny than used in a PCA because for example, the proximity of smuts, yeasts and zygomycetes is probably due to convergent losses not to their intermediate evolutionary position. Proximity can't be interpreted as indicating evolutionary intermediates.

Answer: We partly disagree with this and think that proximity, in this analysis, adequately reflects gene content similarity between groups. We don't think that simply convergent loss can explain the proximity of the highlighted clades without a substantial number of shared HGs. For example, Microsporidia, which exhibit excessive genome reduction, rather than grouping with other simplified species (such as yeasts), formed a distinct group. See the Microsporidia species highlighted with blue ellipse in the figure below. Regarding patterns of the retention of homologs, please refer to Figure 2.

Figures and conclusions

Figure 1. The PCA should be uncluttered by using letters, not circles to more clearly designate clade membership. Even zooming in to look at the high resolution download, it's hard to see which data points are from which clades. Remove the icons because they're available just to the right, and removing the dotted lines. The shading in the polygons is too bright and again, it makes the points hard to see.

Answer: As another visualisation, we created a clarified version of the PCoA plot, as Supplementary Fig. 2. Nevertheless, we think that the original version of this plot is more informative for the main figure.

Conclusions will benefit from being tightened up. At present, they do not highlight the best supported aspects of the paper but are a collection of inadequately supported statements.

Answer: We have revised and improved the conclusions section. As mentioned also by another referee, there are a lot of new results in this paper that need to be summarised in the conclusions section, we think we highlighted the most important aspects. These include the similarity of non-dikarya fungi to unicellular opisthokonts, the lack of synapomorphies, loss of 'protist' genes, bursts of duplication and the dominant functional signals therein. We worked on the Conclusions so that these points are arranged in a better storyline now.

Figure 3 shows the same pattern of change in holozoa and metazoa as in fungal phyla. What's the basis for arguing for a contrast in patterns in different kingdoms?

327 content has changed drastically during early fungal evolution. However, in contrast to animals³ and plants², this has not happened abruptly at the taxonomic limit of

the kingdom, even if competing 329 taxonomic circumscriptions of Fungi are considered.

Answer: We are not sure what pattern of change the Reviewer is referring to here, but the quoted section synthesises multiple observations, not only those related to gene duplication and loss, which Figure 3 shows. In this study, we did not intend to compare the dynamics of changes in homologous groups between Holozoan and Holomycotan lineages, since this was recently done by Ocaña-Pallarès et al. Nature (2022), but we have focused on changes in the gene repertoire in the Holomycotan lineage. Therefore, our dataset is not sufficient to draw conclusions about gene repertoire changes along the Holozoan lineage. In the highlighted sentence and paragraph (L387-390), we discussed the changes in protein-coding gene content (emergence and loss of entire HGs shown in Fig 2), and we relied on literature data in the case of plants (Bowles et al. Curr. Biol. 2020) and animals (Fernández & Gabaldón Nat. Ecol. Evol. 2020). The contrast we highlighted is that the taxonomic border of both plants and animals coincides with significant genetic change, whereas in fungi this is not the case.

Line 355 Wording needs re-consideration 'in which gene loss aided the gradual shedding of protist traits' Exactly which genes and traits are meant here and what evidence suggests that gene loss helped lose ancestral traits? Doesn't gene loss usually follow loss of selection for function, as with the flagellar apparatus, where gene loss seems to be quite quick after flagellar apparatus (an ancestral structure) was lost from various lineages. Ophisthospordia underwent drastic genome reduction, not gradual gene loss.

Answer: We agree and have replaced "aided" with "followed".

Line 378. Not much evidence of 'a unicellular algal parasite ancestor'. Nucleariids seem able to digest bacteria and assorted microbes and Opisthospordia are endoparasites. With the diversity among extant taxa that diverged from early within fungi, it's unreasonable to assume that the ancestral fungus was aphelid-like.

Answer: We have replaced "unicellular algal parasite" by "Opisthokont ancestor" in this sentence. However, we note that several recent results point out that a unicellular algal parasite is a likely ancestral state for the fungi (Galindo et al. 2021 Nat. Comms; Chang et al. 2015 Genome Biol. Evol.)

Fig. S3. Explain P/F

Answer: We corrected it to N/F and explained it in detail in the legend.

Reviewer #3 (Remarks to the Author):

Merenyi et al have profiled the genomic content of 123 fungi and related eukaryotes to attempt to chart evolutionary trends within the fungal kingdom. The authors find that the evolution of Fungi arising from Opisthokonts and their subsequent genomic development was episodic in nature, marked by gradual turnover of “protistan” gene content across all major fungal lineages and more interspersed events of probable gene family and/or genome duplication. This turnover appears to be more pronounced among genes related to transport and transcription processes. Compared to other eukaryote groups of similar rank, such as Metazoa, Fungi appear to lack distinct synapomorphies that would be commonly used to distinguish other major eukaryote nodes. Taken as a whole, the authors suggest that there is some dissonance between how we classify "Fungi" taxonomically and their relative lack of synapomorphies at the genomic level. This seems a reasonable conclusion given the fact that this type of analysis encompasses millions of years of different evolutionary processes and fungi have evolved very differently from plants or animals.

I think this manuscript is of appreciable quality to anyone studying fungal or eukaryote genome evolution and would expect to see this kind of work in a journal like NEE. The approach and methodology outlined in this study would also be suited to other tricky parts of the eukaryote tree. I have some suggestions below which hopefully will improve upon the manuscript in its current state, but otherwise I would recommend this for publication pending some revisions.

Answer: Thank you for the words of appreciation.

Comments:

Overall there are several typological errors in this manuscript such commas being in places they don't need to be in, and some sentences that could be rephrased or broken up to improve readability. I think this would not require much work outside of simply reading over the manuscript once or twice. Otherwise, the figures and supplementary information provided are clear and easy to understand.

Answer: We double checked the punctuation and the text, we think this aspect has also improved a lot in the revision.

Introduction

Pg 1, Lines 38: Not sure the “conversely” is needed here, this is referring to two notions (identifying the genomic divergence of a new lineage from its relative and the subsequent innovations within that lineage) that aren’t necessarily contrasting in my view.

Answer: Thank you. We corrected it.

Pg 2, Lines 46-61: It may be worth rephrasing this at the start to contrast the general division of Fungi into the (generally) sessile osmotrophs of the Dikarya, which is the part of “Fungi” most readers will probably be familiar with, and the EDF which are harder to distinguish from protists and would need outlining. This can then lead into the discussion of LUFA. At the moment, the mention of Dikarya comes at the end.

Pg 2, Lines 59-60: “However” appears in this sentence twice.

Answer: We have rephrased this paragraph, in order to mention Dikarya before the non-Dikarya fungi (L51-L53).

Results

Pg 2, Line 75: Unnecessary commas such as “all, but one”. Is Sanchythriomycota a typo?

Answer: Thank you. We corrected them.

Pg 3, Figure 1b: If possible, I’d appreciate if the authors could include the full PCoA plot with species labelling in the supplementary material like they have with the phylogenetic tree. For me this is the most striking figure in the manuscript, and it would be interesting to see internal relationships within those phylum clusters. For example, I’d be curious to know if the broad-looking distribution of the chytrids in the PCoA plot is related to the divide between the Neocallimastigo- group and other chytrids.

Answer: The suggested plot was added as Supplementary Fig. 2. Indeed, Neocallimastigomycota is separated from other Chytrids.

21Pg 3, Line 104-111: This is presented in a slightly confusing manner. There are 540 HGs that are absent in Dikarya but are present in other all sampled Holomycota (EDF+Nucleariids in this case). LUFA lost 41 of these when it diverged from its closest sampled relatives, the Nucleariids. I would rephrase this paragraph. In Figure 2, this number is given as 31.

Answer: We have rephrased this paragraph (L122-131). For better clarity, we have added another explanation in Fig. 2, because 31 is the number of novel fungal HGs.

Pg 3, Line 119: The authors mention later on (Line 131) that GO annotation is more precise in metazoans than in Fungi. One could argue that the relative bias of GO annotations towards human and model organism data creates an artificial limit on what we can say about their distribution in non-model organisms. Arguably, the authors observed this themselves by the number of unannotated HGs they found (Lines 155-157). I think the authors could highlight the limitations of GO enrichment analysis as is

currently implemented in this regard, even though it is a perfectly reasonable analysis to perform otherwise.

Answer: We clarified, and emphasised the limitations of this approach in L162-165, and also in the methods section L518-520.

Pg 4, Line 124-5: Some typos, I would use “despite” instead of “despite of” and “between” instead of “among”.

Answer: Thank you, we corrected them.

Pg 4, Line 128: The authors use the term "RM clade" here to refer to Rozellomycota+Microsporidia, but in the Introduction and Conclusion they use the term "Opisthosporidia".

Answer: We corrected it.

Pg 4, Line 142: There’s a typo in the citation here, “see 4th chapter”?

Answer: We replaced it with “see below”.

When the authors refer to the "split of chytrids" as they do later in the manuscript, it's not immediately obvious as to whether they mean the divergence of the chytrids from other Fungi or to internal divergence within the chytrids.

Answer: We mean split of xy from other fungi and rephrased the relevant section in the ms accordingly.

Pg 6, Lines 218-219: My understanding is that the Neocalli- group are almost exclusively comprised of species found in the gut of ruminants. I don't know whether that environmental limitation has anything to do with the lineage-specific duplications seen in Figure 3, or whether those environments are any more conducive to bacteria-fungi HGT than e.g. soil. The manuscript shies away from discussing HGT otherwise, which is fine, but raising the subject here may require some additional explanation.

Answer: We have not attempted to analyse the HGT events separately, as there was no significant amount of HGT in the period of early fungal evolution we have studied, based

on our unpublished data and what is documented in the literature. Therefore, we have relied on the literature here, which we have emphasised in L256-258.

Conclusion

No comments.

Methods

Pg 10, Line 382: I would include the reference for Sanchytriomycota (I presume this is Galindo et al 2021 in Nature Communications).

Pg 11, Line 434: I would include the word "analysis" after the mention of InterProScan.

Answer: Thank you, added both.

Decision Letter, first revision:

18th April 2023

Dear László,

Thank you for submitting your revised manuscript "Taxonomic vs genomic fungi: contrasting evolutionary loss of ancestral genomic heritage and punctuated emergence of fungal novelties" (NATECOLEVOL-221218118A). It has now been seen again by the original reviewers and their comments are below. The reviewers find that the paper has improved in revision, and therefore we'll be happy in principle to publish it in Nature Ecology & Evolution, pending minor revisions to satisfy the reviewers' final requests and to comply with our editorial and formatting guidelines.

[REDACTED]

Reviewer #1 (Remarks to the Author):

The authors have addressed all my comments and I consider that the manuscript is now ready for its

24publication. I congratulate the authors for a great piece of scientific research.

Reviewer #2 (Remarks to the Author):

The revisions and rebuttal effectively addressed some criticism. The authors ruled out that branch lengths were correlated with HG group gain and loss and the language is much clearer.

However, the revised version would be more convincing if the interpretations of the data seemed fully dispassionate.

As in the authors' interpretation, fungi may be hard to define because they retain many ancestral genes. However, another possibility is that fungi are hard to define because both ancestral and derived, fungal-specific characters were lost in early lineages. The extent of early gene loss is a fascinating finding in this paper, but it throws a wrench into ancestral reconstruction, which must be taken into account.

Large gene loss events occurred in lineages that diverged early e.g. at the base of opisthosporidia, 7724 genes are lost. In the ancestor of Chytridiomycota, 4143 genes are lost. The extensive gene losses among lineages limits our power to reliably estimate ancestral gene complements. Not only ancestral protist genes, but early fungal specific genes were likely also lost.

1. I respectfully disagree with the authors' interpretation of PCoA as showing an intermediate evolutionary HG complement in non-dikarya fungi. The PCoA draws on a distance matrix of presence and absence of HG. Presence is based on evidence of homology. Absence carries no such evidence. Fig. 3 shows large numbers of gene losses in lineages that diverged early from Dikarya and convergent gene loss is a plausible explanation for phylum similarity in PCoA. The PCoA in Fig 1 should not be considered a picture of ancestral gene complement.

That microsporidia are separated doesn't help (as argued in the rebuttal) because microsporidia not only have a reduced genome, they have a highly divergent genome.

2. The following points are still centered around origins of Dikarya and not the overall pattern of fungal evolution:-

'The largest loss events, 76, 64, and 239 HGs, were inferred in nodes where 110 Blastocladiomycota, Zoopagomycota, and Dikarya, respectively, split from their immediate ancestors.'

Also

'This pattern suggests that genes conserved in pre-fungal protists were lost in a stepwise manner, and non-Dikarya fungi possesses a substantial number of HGs shared with non-fungal lineages, considerably more than previous anecdotal evidence suggested.'

25Also

Taken together, our analyses revealed that non-Dikarya fungi possess a large number of HGs 161 shared specifically with protists, and that these were gradually lost during evolution.

3. Ocaña-Pallarès et al. Nature (2022), (a paper mentioned in the rebuttal) should be discussed here, as similarities and differences in results concerning evolution of fungal HG groups are directly relevant to this manuscript.

Line 196. 'We find that gene duplication has been highly episodic'. The mapping of duplications contrasts with results of a similar analysis (Ocaña-Pallarès et al. Nature (2022) Extended data Fig. 10). What's the explanation for the difference?

4. Figure 2. What is the basis for HG selection? Why 540 in the ancestor and 0 in Dikarya? Dikarya share many genes with metazoans, so the the complete loss of all ancestral HG doesn't make sense and explanation is needed.

Minor points:

5. classification line 76. Fungi are also in Amorphea, so add 'other' as in:

"we selected representatives of all but one currently accepted

76 Holomycota phyla (except Sanchytriomycota), as well as OTHER Amorphea"

6. Methods and intro. Defining 'protist' is a good approach but specify which ancestral clade is meant to be used. Unicellular opisthokonts? Unicellular Amorphea?

7. Line 105 and in methods. Clarify. 70% of what is conserved among which group? (In showing '> 70% conservation').

8. Specify the node(s) where unannotated proteins arise:

"Finally, HGs containing unannotated proteins are prevalent (17.8% of 163 and 14.5% of 186 HGs) 159 among core fungal novelties, highlighting the understudied status of fungal-specific genes."

9. RM vs Opisthosporidia still inconsistent in text (eg line 130) and supplements.

10. Clarify, does "four proteins" refer to an HG, with a minimum of four paralogs, or is the gene family a set of orthologs present in at least four taxa?

"433 For gene tree reconstructions, gene families containing at least four proteins were aligned"

11. Fig. 2 legend. Order of abbreviations should be organized for reader convenience. Could be alphabetical or in order of genomic events.

12. If possible, give supplement files informative names. Which file, for example, is supplementary table 1? A ton of work went into supplement and they're cool, so they should be as accessible as possible.

26Here's what the names currently look like in a downloads folder:

19240_1_data_set_193836_rs0pk2.xlsx
19240_1_data_set_193837_rs0pk2.xlsx
19240_1_data_set_193830_rs0pk2.xlsx

Reviewer #3 (Remarks to the Author):

I think the revised manuscript from Merenyi et al is suitable for publication. I have some small typo/grammar comments, the authors can address or ignore them if they wish. Great work overall.

Comments:

Pages 1-2, Lines 42-44: I would rephrase this to "They exhibit extreme diversity in both morphology and ecological function, and play key roles in many ecosystems as symbionts, parasites and saprobes among others".

Page 2, Line 45: I would put the septate and thalli remarks as an example within parentheses - "(e.g. non-motile septate)" etc.

Page 9, Line 345: "Noteworthy" instead of "remarkable"?

Page 10, Line 383: "Although" instead of "albeit"?

Our ref: NATECOLEVOL-221218118A

24th April 2023

Dear Dr. Nagy,

Thank you for your patience as we've prepared the guidelines for final submission of your Nature Ecology & Evolution manuscript, "Taxonomic vs genomic fungi: contrasting evolutionary loss of ancestral genomic heritage and punctuated emergence of fungal novelties" (NATECOLEVOL-221218118A). Please carefully follow the step-by-step instructions provided in the attached file, and add a response in each row of the table to indicate the changes that you have made. Please also check and comment on any additional marked-up edits we have proposed within the text. Ensuring that each point is addressed will help to ensure that your revised manuscript can be swiftly handed over to our

27production team.

****We would like to start working on your revised paper, with all of the requested files and forms, as soon as possible (preferably within two weeks). Please get in contact with us immediately if you anticipate it taking more than two weeks to submit these revised files.****

In recognition of the time and expertise our reviewers provide to Nature Ecology & Evolution's editorial process, we would like to formally acknowledge their contribution to the external peer review of your manuscript entitled "Taxonomic vs genomic fungi: contrasting evolutionary loss of ancestral genomic heritage and punctuated emergence of fungal novelties". For those reviewers who give their assent, we will be publishing their names alongside the published article.

Nature Ecology & Evolution offers a Transparent Peer Review option for new original research manuscripts submitted after December 1st, 2019. As part of this initiative, we encourage our authors to support increased transparency into the peer review process by agreeing to have the reviewer comments, author rebuttal letters, and editorial decision letters published as a Supplementary item. When you submit your final files please clearly state in your cover letter whether or not you would like to participate in this initiative. Please note that failure to state your preference will result in delays in accepting your manuscript for publication.

Cover suggestions

As you prepare your final files we encourage you to consider whether you have any images or illustrations that may be appropriate for use on the cover of Nature Ecology & Evolution.

28Nature Ecology & Evolution has now transitioned to a unified Rights Collection system which will allow our Author Services team to quickly and easily collect the rights and permissions required to publish your work. Approximately 10 days after your paper is formally accepted, you will receive an email in providing you with a link to complete the grant of rights. If your paper is eligible for Open Access, our Author Services team will also be in touch regarding any additional information that may be required to arrange payment for your article.

Please note that *Nature Ecology & Evolution* is a Transformative Journal (TJ). Authors may publish their research with us through the traditional subscription access route or make their paper immediately open access through payment of an article-processing charge (APC). Authors will not be required to make a final decision about access to their article until it has been accepted. [Find out more about Transformative Journals](https://www.springernature.com/gp/open-research/transformative-journals)

Authors may need to take specific actions to achieve [compliance with funder and institutional open access mandates](https://www.springernature.com/gp/open-research/funding/policy-compliance-faqs). If your research is supported by a funder that requires immediate open access (e.g. according to [Plan S principles](https://www.springernature.com/gp/open-research/plan-s-compliance)) then you should select the gold OA route, and we will direct you to the compliant route where possible. For authors selecting the subscription publication route, the journal's standard licensing terms will need to be accepted, including [a href="https://www.nature.com/nature-portfolio/editorial-policies/self-archiving-and-license-to-publish">those licensing terms will supersede any other terms that the author or any third party may assert apply to any version of the manuscript](https://www.nature.com/nature-portfolio/editorial-policies/self-archiving-and-license-to-publish).

[REDACTED]

[REDACTED]

Reviewer #1:

Remarks to the Author:

The authors have addressed all my comments and I consider that the manuscript is now ready for its publication. I congratulate the authors for a great piece of scientific research.

Reviewer #2:

Remarks to the Author:

The revisions and rebuttal effectively addressed some criticism. The authors ruled out that branch lengths were correlated with HG group gain and loss and the language is much clearer.

However, the revised version would be more convincing if the interpretations of the data seemed fully dispassionate.

As in the authors' interpretation, fungi may be hard to define because they retain many ancestral genes. However, another possibility is that fungi are hard to define because both ancestral and derived, fungal-specific characters were lost in early lineages. The extent of early gene loss is a fascinating finding in this paper, but it throws a wrench into ancestral reconstruction, which must be taken into account.

Large gene loss events occurred in lineages that diverged early e.g. at the base of opisthosporidia, 7724 genes are lost. In the ancestor of Chytridiomycota, 4143 genes are lost. The extensive gene losses among lineages limits our power to reliably estimate ancestral gene complements. Not only ancestral protist genes, but early fungal specific genes were likely also lost.

1. I respectfully disagree with the authors' interpretation of PCoA as showing an intermediate evolutionary HG complement in non-dikarya fungi. The PCoA draws on a distance matrix of presence and absence of HG. Presence is based on evidence of homology. Absence carries no such evidence. Fig. 3 shows large numbers of gene losses in lineages that diverged early from Dikarya and convergent gene loss is a plausible explanation for phylum similarity in PCoA. The PCoA in Fig 1 should not be considered a picture of ancestral gene complement.

That microsporidia are separated doesn't help (as argued in the rebuttal) because microsporidia not only have a reduced genome, they have a highly divergent genome.

2. The following points are still centered around origins of Dikarya and not the overall pattern of fungal evolution:-

'The largest loss events, 76, 64, and 239 HGs, were inferred in nodes where 110 Blastocladiomycota, Zoopagomycota, and Dikarya, respectively, split from their immediate ancestors.'

30Also

'This pattern suggests that genes conserved in pre-fungal protists were lost in a stepwise manner, and non-Dikarya fungi possesses a substantial number of HGs shared with non-fungal lineages, considerably more than previous anecdotal evidence suggested.'

Also

Taken together, our analyses revealed that non-Dikarya fungi possess a large number of HGs 161 shared specifically with protists, and that these were gradually lost during evolution.

3. Ocaña-Pallarès et al. Nature (2022), (a paper mentioned in the rebuttal) should be discussed here, as similarities and differences in results concerning evolution of fungal HG groups are directly relevant to this manuscript.

Line 196. 'We find that gene duplication has been highly episodic'. The mapping of duplications contrasts with results of a similar analysis (Ocaña-Pallarès et al. Nature (2022) Extended data Fig. 10). What's the explanation for the difference?

4. Figure 2. What is the basis for HG selection? Why 540 in the ancestor and 0 in Dikarya? Dikarya share many genes with metazoans, so the the complete loss of all ancestral HG doesn't make sense and explanation is needed.

Minor points:

5. classification line 76. Fungi are also in Amorphea, so add 'other' as in:

"we selected representatives of all but one currently accepted

76 Holomycota phyla (except Sanchytriomycota), as well as OTHER Amorphea"

6. Methods and intro. Defining 'protist' is a good approach but specify which ancestral clade is meant to be used. Unicellular opisthokonts? Unicellular Amorphea?

7. Line 105 and in methods. Clarify. 70% of what is conserved among which group? (In showing '> 70% conservation').

8. Specify the node(s) where unannotated proteins arise:

"Finally, HGs containing unannotated proteins are prevalent (17.8% of 163 and 14.5% of 186 HGs) 159 among core fungal novelties, highlighting the understudied status of fungal-specific genes."

9. RM vs Opisthosporidia still inconsistent in text (eg line 130) and supplements.

10. Clarify, does "four proteins" refer to an HG, with a minimum of four paralogs, or is the gene family a set of orthologs present in at least four taxa?

"433 For gene tree reconstructions, gene families containing at least four proteins were aligned"

11. Fig. 2 legend. Order of abbreviations should be organized for reader convenience. Could be

31alphabetical or in order of genomic events.

12. If possible, give supplement files informative names. Which file, for example, is supplementary table 1? A ton of work went into supplement and they're cool, so they should be as accessible as possible.

Here's what the names currently look like in a downloads folder:

19240_1_data_set_193836_rs0pk2.xlsx
19240_1_data_set_193837_rs0pk2.xlsx
19240_1_data_set_193830_rs0pk2.xlsx

Reviewer #3:

Remarks to the Author:

I think the revised manuscript from Merenyi et al is suitable for publication. I have some small typo/grammar comments, the authors can address or ignore them if they wish. Great work overall.

Comments:

Pages 1-2, Lines 42-44: I would rephrase this to "They exhibit extreme diversity in both morphology and ecological function, and play key roles in many ecosystems as symbionts, parasites and saprobes among others".

Page 2, Line 45: I would put the septate and thalli remarks as an example within parentheses - "(e.g. non-motile septate)" etc.

Page 9, Line 345: "Noteworthy" instead of "remarkable"?

Page 10, Line 383: "Although" instead of "albeit"?

Final Decision Letter:

11th May 2023

Dear László,

We are pleased to inform you that your Article entitled "Genomes of fungi and relatives reveal delayed loss of ancestral gene families and evolution of key fungal traits", has now been accepted for publication in Nature Ecology & Evolution.

Over the next few weeks, your paper will be copyedited to ensure that it conforms to Nature Ecology and Evolution style. Once your paper is typeset, you will receive an email with a link to choose the appropriate publishing options for your paper and our Author Services team will be in touch regarding any additional information that may be required

After the grant of rights is completed, you will receive a link to your electronic proof via email with a

32request to make any corrections within 48 hours. If, when you receive your proof, you cannot meet this deadline, please inform us at rjsproduction@springernature.com immediately.

You will not receive your proofs until the publishing agreement has been received through our system

Due to the importance of these deadlines, we ask you please us know now whether you will be difficult to contact over the next month. If this is the case, we ask you provide us with the contact information (email, phone and fax) of someone who will be able to check the proofs on your behalf, and who will be available to address any last-minute problems . Once your paper has been scheduled for online publication, the Nature press office will be in touch to confirm the details.

Acceptance of your manuscript is conditional on all authors' agreement with our publication policies (see www.nature.com/authors/policies/index.html). In particular your manuscript must not be published elsewhere and there must be no announcement of the work to any media outlet until the publication date (the day on which it is uploaded onto our web site).

Please note that *Nature Ecology & Evolution* is a Transformative Journal (TJ). Authors may publish their research with us through the traditional subscription access route or make their paper immediately open access through payment of an article-processing charge (APC). Authors will not be required to make a final decision about access to their article until it has been accepted. [Find out more about Transformative Journals](https://www.springernature.com/gp/open-research/transformative-journals)

Authors may need to take specific actions to achieve [compliance](https://www.springernature.com/gp/open-research/funding/policy-compliance-faqs) with funder and institutional open access mandates. If your research is supported by a funder that requires immediate open access (e.g. according to [Plan S principles](https://www.springernature.com/gp/open-research/plan-s-compliance)) then you should select the gold OA route, and we will direct you to the compliant route where possible. For authors selecting the subscription publication route, the journal's standard licensing terms will need to be accepted, including [those licensing terms](https://www.nature.com/nature-portfolio/editorial-policies/self-archiving-and-license-to-publish) will supersede any other terms that the author or any third party may assert apply to any version of the manuscript.

An online order form for reprints of your paper is available at <https://www.nature.com/reprints/author->

reprints.html"><https://www.nature.com/reprints/author-reprints.html>. All co-authors, authors' institutions and authors' funding agencies can order reprints using the form appropriate to their geographical region.

We welcome the submission of potential cover material (including a short caption of around 40 words) related to your manuscript; suggestions should be sent to Nature Ecology & Evolution as electronic files (the image should be 300 dpi at 210 x 297 mm in either TIFF or JPEG format). Please note that such pictures should be selected more for their aesthetic appeal than for their scientific content, and that colour images work better than black and white or grayscale images. Please do not try to design a cover with the Nature Ecology & Evolution logo etc., and please do not submit composites of images related to your work. I am sure you will understand that we cannot make any promise as to whether any of your suggestions might be selected for the cover of the journal.

You can generate the link yourself when you receive your article DOI by entering it here: http://authors.springernature.com/share.

[REDACTED]

P.S. Click on the following link if you would like to recommend Nature Ecology & Evolution to your librarian <http://www.nature.com/subscriptions/recommend.html#forms>

** Visit the Springer Nature Editorial and Publishing website at www.springernature.com/editorial-and-publishing-jobs for more information about our career opportunities. If you have any questions please click here.**